# SoftCAM: Making black box models self-explainable for medical image analysis

**Kerol Djoumessi**[1] [ID]                    KEROL.DJOUMESSI-DONTEU@UNI-TUEBINGEN.DE
[1] *Hertie Institute for AI in Brain Health, University of Tübingen, Germany*

**Philipp Berens**[1,2] [ID]                    PHILIPP.BERENS@UNI-TUEBINGEN.DE
[2] *Tübingen AI Center, University of Tübingen, Germany*

**Editors:** Accepted for publication at MIDL 2026

## Abstract

Convolutional neural networks (CNNs) are widely used for high-stakes applications like medicine, often surpassing human performance. However, most explanation methods rely on post-hoc attribution, approximating the decision-making process of already trained black-box models. These methods are often sensitive, unreliable, and fail to reflect true model reasoning, limiting their trustworthiness in critical applications. In this work, we introduce SoftCAM, a straightforward yet effective approach that makes standard CNN architectures inherently interpretable. By removing the global average pooling layer and replacing the fully connected classification layer with a convolution-based class evidence layer, SoftCAM preserves spatial information and produces explicit class activation maps that form the basis of the model's predictions. Evaluated on three medical datasets spanning three imaging modalities, SoftCAM maintains classification performance while significantly improving both the qualitative and quantitative explanation compared to existing post-hoc methods. The code is available at https://github.com/kdjoumessi/SoftCAM.

**Keywords:** ElasticNet, Self-explainable models, Convolutional Neural Networks, Class Activation Maps, Attribution maps.

## 1. Introduction

Convolutional Neural Networks (CNNs) have revolutionized computer vision by effectively capturing local spatial patterns, reducing the number of parameters, and enabling faster convergence, leading to state-of-the-art performance in tasks such as image recognition and object detection (Oquab et al., 2015; Li et al., 2021). However, their lack of interpretability limits adoption in high-stakes fields such as medical image analysis, where transparency and trust are essential. To mitigate this issue, numerous post-hoc saliency- or attribution-based methods have been proposed to explain CNN predictions by highlighting "where" important features appear in the input through pixel-wise importance maps computed after training. Prominent approaches include class activation maps (CAM) techniques (Zhou et al., 2016; Selvaraju et al., 2017; He et al., 2022), which combine convolutional features with class-specific weights; backpropagation-based methods (Springenberg et al., 2014; Sundararajan et al., 2017), which propagate output gradients to the input to estimate pixel relevance; and perturbation-based methods (Wang et al., 2020; Ivanovs et al., 2021), which systematically alter input features and measure the corresponding change in the model's output.

While post-hoc attribution-based methods provide intuitive visualizations of discriminative regions, they frequently approximate rather than accurately reflect the model's true reasoning (Adebayo et al., 2018; Saporta et al., 2022) and are often valid only under strong regularity assumptions (Günther et al., 2025). Their post-hoc nature further limits their effectiveness—particularly in clinical applications (Arun et al., 2021)—due to low faithfulness, reliability, and consistency. Consequently, their visual outputs may not accurately reflect the model's internal decision-making (Adebayo et al., 2018; Saporta et al., 2022). Moreover, post-hoc techniques often struggle to precisely localize disease-relevant regions in medical images (Arun et al., 2021), where the scarcity of ground-truth annotations further complicates validation. To address these challenges, inherently or self-explainable models have been proposed (Rudin, 2019), embedding interpretability directly into their architectures (Brendel and Bethge, 2019; Chen et al., 2019; Koh et al., 2020; Djoumessi et al., 2024a). By coupling prediction with explanation, self-explainable models provide interpretable and transparent insights, in line with human reasoning. However, their reliance on specialized architectures limits generalization to widely used CNN models.

Motivated by these limitations, we introduce SoftCAM, a simple yet effective framework that makes CNNs inherently interpretable without relying on post-hoc explanation methods. SoftCAM generalizes the concept of class activation maps to transform conventional CNNs into self-explainable models. By removing the final global average pooling layer and replacing the fully connected classifier with a convolution-based class-evidence layer, Soft-CAM converts standard CNNs into fully convolutional architectures that generate explicit, class-specific evidence maps used both for prediction and visual explanation. Evaluated on two widely used CNN architectures, we showed that the resulting SoftCAM-based models maintain competitive accuracy relative to their black-box baselines while achieving superior interpretability across three clinically relevant medical imaging datasets spanning three modalities. Furthermore, applying ElasticNet regularization, which combines ridge and lasso penalties, to the evidence maps improves explanations both qualitatively and quantitatively, revealing a task-dependent trade-off between sparsity and density. In addition, we introduce a novel explainability metric, *activation sensitivity*, which penalizes false negatives and weak activations within expert-annotated regions. Finally, a comprehensive evaluation against six widely used post-hoc attribution methods demonstrated that SoftCAM consistently outperforms these approaches on various explainability metrics.

## 2. Method

**Preliminaries** Given an input image $\mathbf{X} \in \mathbb{R}^{H_X \times W_X \times C_X}$ with height $H_X$, width $W_X$, and the number of channels $C_X$, consider a CNN network $f_\theta$ that maps $\mathbf{X}$ to a probability distribution over $C$ classes, $\hat{\mathbf{y}} = f_\theta(\mathbf{X}) \in \mathbb{R}^C$, where $y^c \in \mathbf{y}$ denotes the predicted probability for class $c$. The network consists of a feature extractor $g_\phi$, and a classifier layer $h_\psi$, with learnable parameters $\phi$ and $\psi$, respectively. The feature extractor produces a feature map $\mathbf{Z} = g_\phi(\mathbf{X}) \in \mathbb{R}^{N \times M \times D}$, where $N \times M$ is the spatial resolution and $D$ is the feature dimension (e.g., $D = 2048$ for standard ResNet variants). The classifier then generates the final prediction based on $\mathbf{Z}$. Let $\mathcal{A} = \{\mathbf{A}_k\}_{k=1}^D$ denote the set of activation maps from the feature extractor, with $A_k \in \mathbb{R}^{N \times M}$ representing the $k$-th channel. The 2D low-resolution saliency map $S_{\text{Map}}^c \in \mathbb{R}^{N \times M}$ provides a visual explanation of the model's prediction for

class $c$. This work focuses on training self-explainable CNN classifiers that simultaneously produce both the class prediction $y^c$ and its corresponding explanation $S^c_{\text{Map}}$.

In contrast, traditional CNNs are black-boxes: a global average pooling layer reduces $\mathbf{Z}$ to a vector of size $1 \times D$, which is then fed into one or more linear fully connected layers (FCL) to generate the final prediction. Post-hoc methods are required to explain decisions.

### 2.1. CAM-based methods

Class Activation Maps (CAM) (Zhou et al., 2016) are closely related to our approach, providing visual explanations of CNN predictions through class-specific saliency maps. CAM operates by linearly combining the final convolutional feature maps with their corresponding importance weights from the FCL classifier to produce class-wise attribution maps:

$$S^c_{\text{CAM}}(x_1, x_2) = \sum_{k=1}^{D} w^c_k A_k(x_1, x_2), \tag{1}$$

where $A_k(x_1, x_2)$ denotes the activation of the $k$ feature map at location $(x_1, x_2)$, and $w^c_k$ is the class-specific importance weight for the $k$ feature map from the fully connected layer.

Originally, CAM was designed for CNNs with a global average pooling (GAP) layer followed by a single FCL. Subsequent extensions, such as GradCAM (Selvaraju et al., 2017) and LayerCAM (Jiang et al., 2021), introduced gradient-based approaches that use gradient to compute importance weights, enabling class-specific explanations for various architectures—particularly those with multiple FCLs after the GAP layer, such as VGG (Simonyan and Zisserman, 2015). In GradCAM, the importance weights are computed by globally averaging the gradients of the target class score as $w^c_k = \frac{1}{N \times M} \sum_i^N \sum_j^N \frac{\partial y^c}{\partial A_k(i,j)}$, where $A_k(i,j)$ represents the activation at spatial location $(i, j)$ in the $k$-th feature map. Following Grad-CAM, several gradient-based and gradient-free extensions have emerged (He et al., 2022). While gradient-based methods differ primarily in how they aggregate gradients to compute importance weights, gradient-free approaches compute weights without backpropagation, often relying on perturbation-based methods like ScoreCAM (Wang et al., 2020).

Despite their widespread clinical use (Ayhan et al., 2022), attribution-based methods have notable limitations (Günther et al., 2025): they provide post-hoc explanations that may not reflect the model's true reasoning (Appendix A). Gradient-based variants can suffer from gradient saturation and false confidence (Wang et al., 2020), while gradient-free approaches are computationally costly, requiring many forward passes on perturbed inputs.

### 2.2. Generalizing CAMs for self-explanability

Motivated by the limitations of post-hoc class activation map-based methods in interpreting CNN, we introduce SoftCAM (Fig. 1), a straightforward modification of black-box CNN classifiers that makes them inherently interpretable. SoftCAM achieves this by replacing the fully connected classification layer in classical CNNs with an explicit class-evidence convolutional layer, preserving spatial information and providing explanations in a single forward pass, eliminating the need and computational overhead for post-hoc techniques.

We make black-box CNN self-explainable by modifying how predictions are obtained. Any FCL of size $b_1 \times b_2$ can be equivalently expressed as a $1 \times 1$ convolutional layer with $b_1$

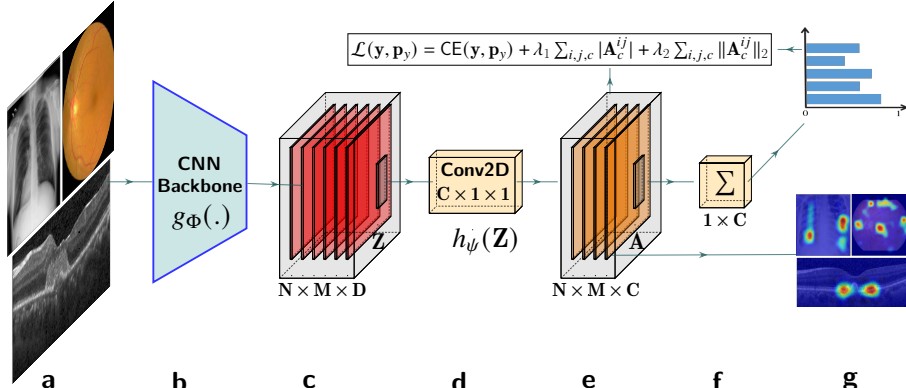

Figure 1: **Overview of SoftCAM architecture.** (**a**) Input image. (**b**) The CNN backbone consists of all layers before the global average pooling layer. (**c**) Feature map generated by the backbone. (**d**) Classifier module with $C$ convolutional kernels of size $1 \times 1$. (**e**) Self-explainable class activation maps **A**, obtained from the classifier with ElasticNet penalty applied to it to enhance interpretability. (**f**) Final predictions are derived from the evidence maps via spatial average pooling followed by the softmax function. Class-specific evidence maps (**g**) are upsampled and overlaid on the input to visualize the model's decision-making process.

input channels and $b_2$ output channels (Donteu et al., 2023). This allows systematic replacement of FCL classifier heads in CNN classifiers with convolutional layers, eliminating the GAP layer before classification while preserving model complexity and spatial localization. The new classifier module $h$ consists of convolutional layers (Fig. 1d) with $C$ convolution kernels of size $1 \times 1$ and unit stride, producing class-specific evidence maps (Fig. 1e):

$$\mathbf{A} = h_\psi(\mathbf{Z}) \in \mathbb{R}^{M \times N \times C}, \tag{2}$$

where $\psi$ is a learnable parameter. These maps can be upsampled to the input resolution and overlaid on the image (Fig. 1g) for visualization and clinical interpretation. The module $h_\psi$ can be viewed as a *soft*, explainable generalization of classical post-hoc attribution methods (Eq. 1), mapping the high-dimensional feature map $\mathbf{Z}$ with $D$ channels into low-dimentional, class-wise activation maps $\mathbf{A}$ with $C$ channels corresponding to the target classes. Unlike CAM (Eq. 1) and related attribution methods (He et al., 2022), our approach leverages the final feature map of the backbone and applies a parameterized function $h_\psi$ to produce class activation maps that directly support prediction. Instead of explicitly defining importance weights, these are implicitly learned and encoded within the classifier's parameters, enabling *soft* generation of class activation maps jointly with prediction during training.

The resulting architecture is a fully convolutional, self-explainable model, where the predicted probabilities are derived directly from the evidence maps (Fig.1e), maintaining the model complexity without adding extra learnable parameters:

$$\hat{\mathbf{y}} = \mathrm{Softmax}\bigg(\mathrm{AvgPool}\Big(h_\psi\big(g_\Phi(\mathbf{X})\big)\Big)\bigg) \in \mathbb{R}^{1 \times C}. \tag{3}$$

Furthermore, the class evidence maps $\mathbf{A}$ serve as built-in explanations, directly representing the contribution of individual input regions to the final prediction (Fig. 1g). Using class-evidence maps for classification offers several advantages: all importance scores are weighted equally when computing class probabilities (Fig. 1f). Consequently, input feature patches with high activations in the evidence maps contribute more strongly to the prediction, analogous to linear models, where each input feature contributes linearly to the output.

## 2.3. Regularizing SoftCAM for interpretability

By using explicit class-evidence maps, SoftCAM-based models can be trained with regularization constraints directly applied to the explanation maps, enhancing interpretability. In practice, we apply an ElasticNet penalty (Zou and Hastie, 2005), which linearly combines the $\ell_1$ (lasso) and $\ell_2$ (ridge) regularization, leading to the following loss function:

$$\mathcal{L}(\mathbf{y}, \hat{\mathbf{y}}) = \mathrm{CE}(\mathbf{y}, \hat{\mathbf{y}}) + \lambda_1 \sum_{i,j,c} |\mathbf{A}_c^{ij}| + \lambda_2 \sum_{i,j,c} ||\mathbf{A}_c^{ij}||_2. \tag{4}$$

Here, CE denotes the cross-entropy loss, and $\mathbf{y}$ represents the reference labels. Setting $\lambda_2 = 0$, results in the lasso penalty promoting sparsity in evidence maps (Donteu et al., 2023) by suppressing less informative activations (mainly false positives), making it particularly useful for tasks where precision in explanations is crucial. In contrast, $\lambda_1 = 0$, gives the ridge penalty, which smooths activations without forcing them to zero, useful for assessing localization sensitivity over large regions, as it penalizes false negatives (Appendix E). ElasticNet thus offers a flexible balance between lasso ($\lambda_2 = 0$) and ridge ($\lambda_1 = 0$) penalties, which can be chosen according to the task and the explainability metric being optimized.

## 3. Experimental setup

**Datasets.** We evaluated our approach on three publicly available medical datasets spanning three imaging modalities: the Kaggle Diabetic Retinopathy (DR) (Dugas et al., 2015), Retinal OCT (Kermany et al., 2018), and the RSNA Chest X-Ray (CXR) (Shih et al., 2019). The first dataset comprised high-resolution retinal color fundus images labeled with DR severity score ranging from 0 (No DR) to 4 (Proliferative DR). The second dataset included retinal OCT B-scans images labeled for three retinal disease conditions. The final dataset consisted of high-resolution frontal-view chest radiographs labeled for pneumonia detection, with bounding boxes for pneumonia cases. Additionally, clinicians annotated lesions in 65 DR images from the Kaggle dataset (Djoumessi et al., 2024b) and 40 drusen lesions from the retinal OCT dataset (Djoumessi et al., 2024a) to allow clinical evaluation. Each dataset was split into training, validation, and test sets, ensuring that all samples from a given patient remained in the same split. For full details, see Appendix B.

**Baseline models.** Our method was evaluated on two widely used black-box CNN architectures: ResNet-50 (He et al., 2016) and VGG-16 (Simonyan and Zisserman, 2015). These models primarily differ in the design of their classification heads: ResNet uses a GAP layer followed by a single linear classifier, whereas VGG flattens the convolutional feature maps and employs multiple fully connected layers for classification. More details on the training setup[2], including data preprocessing and augmentation, are provided in the Appendix C.

Table 1: Classification performance for disease detection on the test sets. SoftCAM variants of both CNNs are denoted by $^{SC}$, with $\ell_\lambda$ indicating the applied penalty.

| | Kaggle Fundus | | | | OCT retinal | | | | RSNA CXR | |
| | Binary | | Multi-class | | Binary | | Multi-class | | Binary | |
| | Acc. | AUC | Acc. | $\kappa$ | Acc. | AUC | Acc. | $\kappa$ | Acc. | AUC |
|---|---|---|---|---|---|---|---|---|---|---|
| VGG-16 | .907 | .938 | **.863** | **.835** | **.994** | **1.0** | **.967** | **.955** | .952 | .989 |
| VGG-16$^{SC}$ | **.915** | **.942** | .861 | .834 | **.994** | **1.0** | .963 | .947 | **.957** | **.999** |
| $\ell_1$-VGG-16$^{SC}$ | .911 | .938 | .859 | .827 | .988 | .999 | .947 | .929 | .953 | .990 |
| $\ell_2$-VGG-16$^{SC}$ | .910 | .937 | .858 | .820 | .988 | .999 | .961 | .946 | .951 | .988 |
| ResNet-50 | .899 | .923 | .850 | .800 | .994 | .999 | .970 | **.963** | **.953** | **.988** |
| ResNet-50$^{SC}$ | .899 | **.926** | .851 | **.811** | .994 | **1.0** | **.974** | .960 | .942 | .983 |
| $\ell_1$-ResNet-50$^{SC}$ | .895 | .923 | .851 | .801 | **.996** | **1.0** | .963 | .955 | .941 | .979 |
| $\ell_2$-ResNet-50$^{SC}$ | **.901** | .924 | **.854** | .810 | .994 | .999 | .965 | .952 | .958 | .986 |

**Post-hoc baseline.** SoftCAM was compared against six widely used post-hoc explanation methods (Appendix D), including CAM-based gradient approaches such as GradCAM (Selvaraju et al., 2017) and LayerCAM (Jiang et al., 2021); CAM-based gradient-free methods such as the original CAM (Zhou et al., 2016) and ScoreCAM (Wang et al., 2020); and backpropagation-based techniques such as Integrated Gradients (Itgd Grad) (Sundararajan et al., 2017) and Guided Backpropagation (Guided BP) (Springenberg et al., 2014). Each post-hoc method was evaluated on its respective black-box CNN models.

**Evaluation metrics.** The models were evaluated for predictive performance and explainability. For explainability, we used several quantitative metrics, including *top-k localization precision* (Donteu et al., 2023); *activation precision* (Barnett et al., 2021), *faithfulness* (Yeh et al., 2019), and *activation consistency* (Donteu et al., 2023), and we further extended activation precision to define *activation sensitivity*. Together, these metrics quantify both alignment with expert clinician annotations and alignment with the model's actual decision-making process. Full metric descriptions are provided in Appendix E.

## 4. Results

### 4.1. Making black box CNNs explainable maintains classification performance

We first evaluated our method on clinically relevant binary classification tasks, including retinal disease classification from color fundus ({0} vs. {1-4}) and OCT retinal images (Normal vs. Drusen), as well as pneumonia detection from chest X-rays, using accuracy and AUC as primary metrics. For each CNN architecture, the "$^{SC}$" model denotes our method without regularization ($\lambda_1 = \lambda_2 = 0$). The "$\ell_1$" and "$\ell_2$" variants are obtained by applying either a lasso ($\lambda_2 = 0$) or a ridge penalty ($\lambda_1 = 0$), respectively, with task- and architecture-specific regularization strenghts (e.g. $\lambda_1 = 1.10^{-6}$ for VGG and $\lambda_1 = 5.10^{-5}$ for ResNet on the fundus dataset). The sparsity hyperparameters were selected to maintain strong classification performance (Appendix F) while yielding qualitatively meaningful visual explanations on a subset of annotated images.

Our results show that SoftCAM-based models (Tab. 1), with explicit class evidence maps, preserve classification performance comparable to their corresponding black-box base-

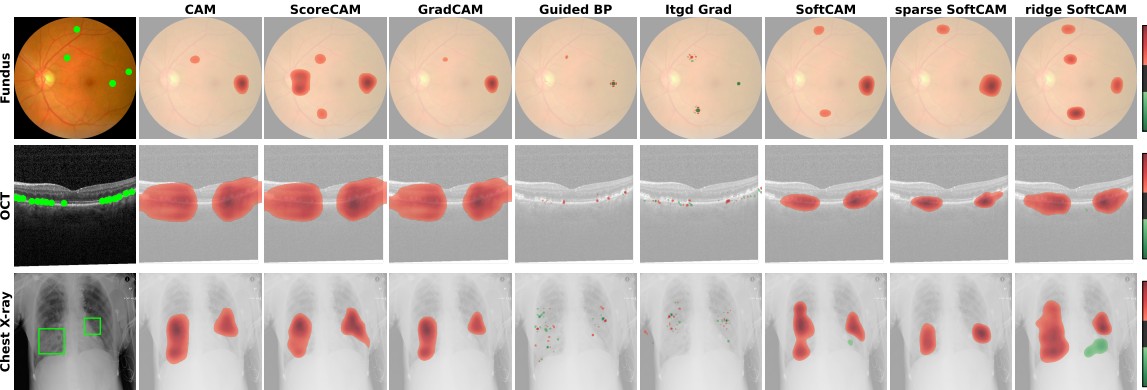

Figure 2: **Example explanations generated by different methods from ResNet-50.**
The first column shows disease images with reference annotations (green markers or bounding boxes). The rows from top to bottom correspond to fundus, OCT, and Chest X-ray images, respectively. The next five columns present saliency maps from different post-hoc explanation methods. The last three columns show our proposed inherently interpretable SoftCAM–based explanations.

lines. Moreover, introducing the lasso and ridge regularizations on the class evidence map did not significantly degrade performance, as is sometimes observed when enforcing interpretability (Rudin, 2019); in some cases, it even led to slight improvements, particularly in retinal disease classification. These findings suggest that using convolutional layers in the classification head is an effective and promising approach for developing high-performing, self-explainable CNN models.

### 4.2. SoftCAM provides inherently interpretable visual explanations

We qualitatively compared the evidence maps of the SoftCAM variants with the saliency maps generated by the six state-of-the-art CAM and back-propagation-based methods. Overall, our method produced more visually interpretable maps with high evidence regions centered on annotated lesions (Fig. 2). We observed that the regions highlighted by the sparse SoftCAM models are typically a subset of those identified by both the unregularized and ridge SoftCAM variants, reflecting the sparsity constraint's effect in reducing irrelevant activations, while ridge regularization promotes denser activations. Additional results, including those for VGG-16 and other examples, are provided in Appendix G.1.

On healthy images, the sparse SoftCAM evidence maps exhibited overall more negative activations, in contrast to the positive activations observed on disease images. To assess this quantitatively, we computed the activation consistency score (Donteu et al., 2023), calculating the proportion of positive and negative activations in disease and healthy samples from the test set. These findings were consistent with qualitative visualization (e.g. sparse SoftCAM vs. SoftCAM on the fundus dataset using ResNet: 0.55 vs. 0.27 for the average proportion of positive activations on disease images). For a full analysis, see Appendix G.2.

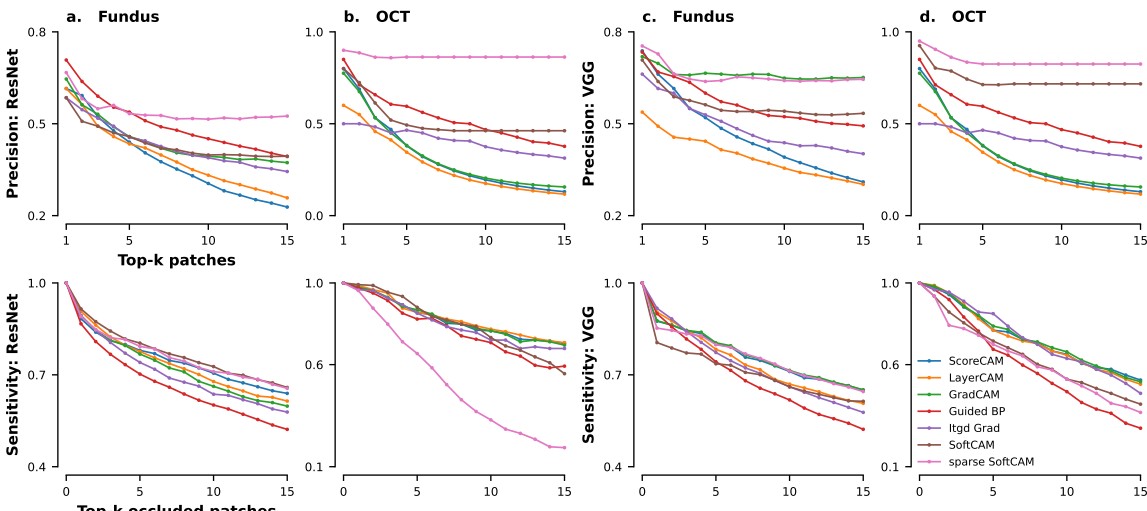

Figure 3: **Quantitative evaluation of explanations methods**. The first row shows the localization precision of saliency maps on fundus and OCT datasets. The second row presents the sensitivity analysis assessing faithfulness. Columns **a,b** show ResNet results, and **c,d** correspond to VGG. Higher precision indicates better localization; lower sensitivity reflects more reliable explanations.

### 4.3. SoftCAM provides localized and faithful explanations

To quantitatively assess the explanations provided by our SoftCAM-based evidence maps compared to post-hoc saliency methods, we first evaluated their localization precision, which measures how consistently the highlighted regions in the explanation maps align with clinician-annotated disease findings. Following (Donteu et al., 2023), we computed the Top-k (k=15) localization precision by upsampling each explanation map to the input resolution, splitting it into non-overlapping $33 \times 33$ patches, and calculating the proportion of positively activated patches that overlap with ground truth annotations. Although inherently interpretable, SoftCAM-based explanations performed competitively overall in terms of localization precision (Fig. 3). Notably, the sparse SoftCAM with the ResNet backbone outperformed all other methods with the highest top-k precision (Appendix G.3 and G.4), and ranked second only in top-3 precision on the fundus dataset (Fig. 3a), behind Guided BP, which benefits from high-resolution saliency maps. Furthermore, the base and sparse SoftCAM typically achieved higher precision with fewer top-k regions, leading to fast convergence, particularly on the Fundus and OCT datasets. This suggests that sparse SoftCAM more consistently highlights fewer, yet truly relevant regions, whereas post-hoc methods produce broader and less specific activations, resulting in higher false-positive rates.

Subsequently, we evaluated the faithfulness (also referred to as sensitivity) of the evidence maps generated by our SoftCAM-based explanations in comparison to post-hoc saliency maps. Sensitivity analysis evaluates how much the highly activated regions in an explanation map contribute to the model's prediction (Yeh et al., 2019), thereby assess-

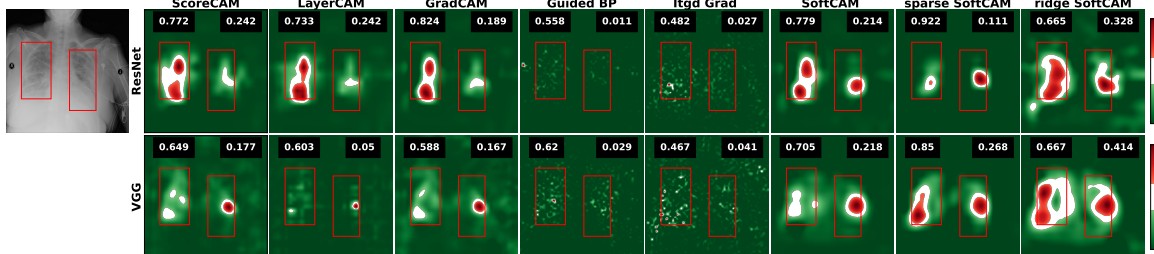

Figure 4: **Example of localization evaluation on the CXR dataset for pneumonia detection**. The first row shows saliency maps generated by different methods from the ResNet model, and the second row from the VGG model. Ground-truth bounding boxes are overlaid on each map, with the top-right value indicating the activation precision, while the top-left value indicates the activation sensitivity.

ing whether the highlighted areas actually influence the model's decision-making process. To do this, we split the input images into non-overlapping $33 \times 33$ patches, then progressively removed the top-ranked patches (based on attribution scores) and measured the relative change in model confidence. We conducted this evaluation on samples that were correctly predicted by both the black-box CNNs and their corresponding SoftCAM variants from the test sets. We found that the sparse SoftCAM generally outperformed other methods, notably on the OCT and RSNA datasets (Fig. 3; Appendix G.3 and G.4). On the fundus dataset, both the base and sparse SoftCAM models performed slightly below the best-performing post-hoc methods, with Guided BP yielding the highest sensitivity scores, followed by Integrated Gradients (Fig. 3). On the OCT dataset, the SoftCAM variants outperformed all post-hoc methods with the ResNet model, while with the VGG the sparse and base versions ranked second and third, respectively. On the RSNA dataset, the sparse SoftCAM achieved the highest sensitivity, surpassing all other methods, with the base SoftCAM ranking second with ResNet and third with VGG (Appendix G.4).

### 4.4. Ridge regularization improves explanation for large disease regions

Since the CXR dataset provided larger bounding boxes localizing disease regions, unlike the sparse point-wise lesion annotations available in the fundus and OCT datasets, we computed activation precision (Barnett et al., 2021), which measures the proportion of the class-guided explanation that falls within the ground-truth bounding boxes, emphasizing precision by penalizing only false positives. However, it does not account for sensitivity by not penalizing false negatives. To address this, we extended this metric by introducing activation sensitivity (Appendix E.3), which penalizes false negatives to better assess the explanation completeness, especially important in clinical imaging tasks where missing relevant regions can be critical (Shih et al., 2019). We further investigated how different regularization methods qualitatively and quantitatively affect explanations. While lasso regularization promotes sparsity by shrinking most false positive activations to zeros, it can lead to suboptimal interpretability on tasks involving large lesion areas. In contrast, ridge

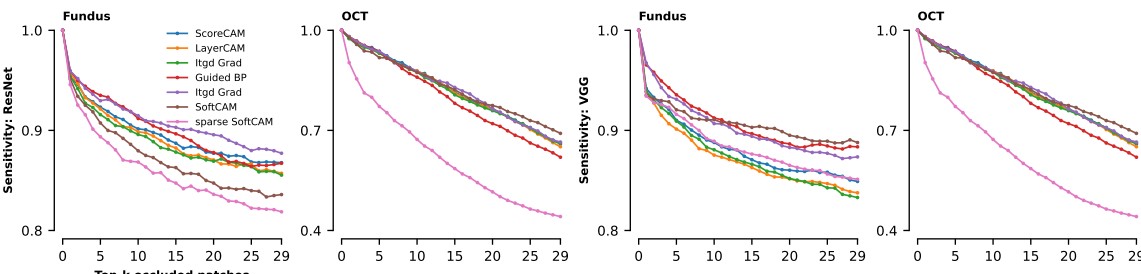

Figure 5: **Sensitivity analysis for the multiclass task.** Evaluation on retinal fundus and OCT datasets assessing how faithfully each explanation method captures the model's internal decision-making process. Lower relative sensitivity indicates more reliable explanations, as they reflect greater changes in output when important features are removed.

regularization encouraged small but nonzero activations, resulting in denser and more informative evidence maps. To evaluate this, we trained a ridge SoftCAM model ($\lambda_1 = 0$) and compared its performance to the sparse SoftCAM, as well as to the post-hoc explanation methods. The ridge regularization strength was selected to balance predictive performance ($\lambda_2 = 7.10^{-5}$ vs. $\lambda_2 = 2.10^{-4}$ for ResNet and VGG; Appendix H.1) while maintaining qualitatively meaningful visual explanations.

Under comparable accuracy (Acc. $\approx 0.95$ for ridge ResNet$^{SC}$, and VGG$^{SC}$), we found that all SoftCAM variants—unregularized, sparse, and ridge—generally outperformed the evaluated post-hoc explanations in both activation precision and activation sensitivity (Fig. 4; Appendix H.2 and H.3). Specifically, sparse SoftCAM achieved the highest activation precision, while ridge SoftCAM excelled in activation sensitivity. The unregularized SoftCAM consistently performed in between, underscoring the importance of balancing lasso and ridge penalties via ElasticNet to suit diverse datasets, tasks, and interpretability requirements; this balance can be selected empirically.

### 4.5. SoftCAM provides faithful explanations for multi-class tasks

Finally, we extended our method to the multi-class setting for retinal disease diagnosis, training the SoftCAM models from ResNet and VGG for DR detection (5 classes, Kaggle dataset) and retinal disease classification (4 classes, OCT dataset). The training setup remained consistent with the binary task, adjusting only the number of classes in the evidence layer and selecting appropriate penalties. Given the small size of retinal lesions, we used lasso regularization, selecting $\lambda_1$ values that preserve predictive performance while providing qualitatively good visualizations on a small set of samples (e.g. $\lambda_1 = 9.10^{-4}$ vs. $\lambda_1 = 3.10^{-6}$ for ResNet and VGG on the OCT dataset; Appendix I.1). Both unregularized and sparse models achieved performance comparable to their respective black-box baselines (Tab. 1), with a slight improvement in Cohen's kappa ($\kappa$) on the fundus dataset when using the ResNet backbone. The quadratic kappa accounts for agreement beyond chance.

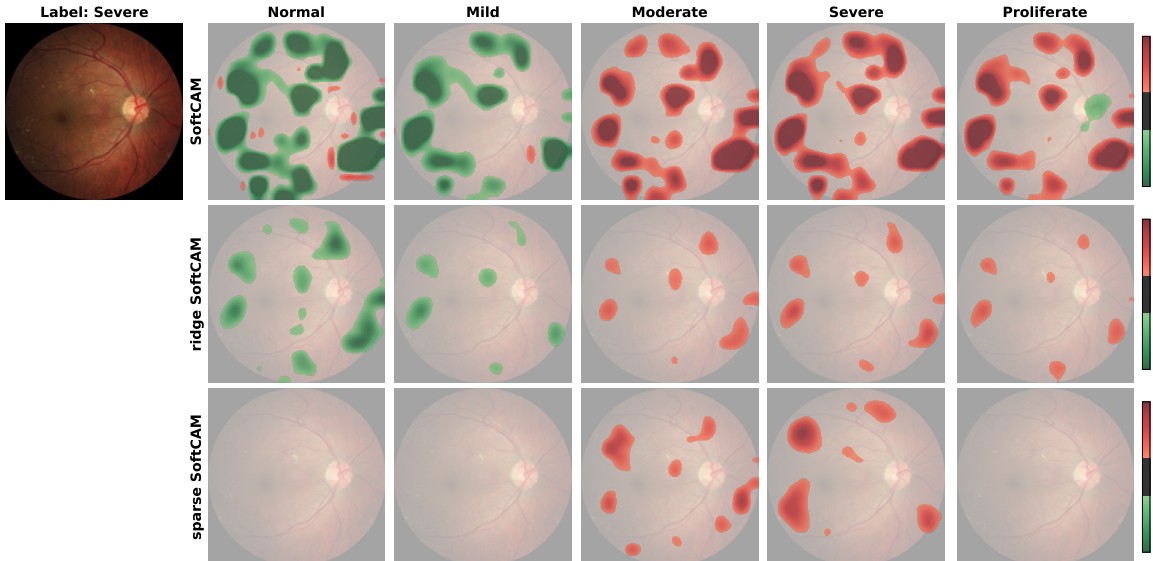

Figure 6: **Examples of multi-class explanations with ResNet.** For a severe diabetic retinopathy case from the Kaggle dataset, the rows show class-specific explanations generated by SoftCAM variants. Integrated Gradients is the best-performing post-hoc method in terms of sensitivity, following the unregularized and sparse SoftCAM variants. The color scale of the explanation maps ranges from green (healthy evidence) to red (disease evidence).

As no ground-truth lesion annotations were available for the multi-class tasks, we evaluated the faithfulness of the explanations by measuring their contribution to model predictions. For correctly classified test samples from all models, we progressively removed top-k ($k = 30$) ranked patches (based on the explanation maps; see Sec.4.3) and tracked the relative average drop in class confidence. In both tasks, sparse SoftCAM achieved the best performance (Fig. 5), yielding the lowest area under the deletion curve and thus the highest faithfulness, except for the VGG backbone on fundus images. Full quantitative results are reported in Appendix I.2.

Notably, the sparse SoftCAM produced class-wise explanations that aligned well with class model confidence, showing minimal evidence in healthy classes (Fig.6; Appendix I.3, for VGG). In the case of DR detection, a progressive disease, it is expected that images labeled with grade $x$, where $1 < x < 5$, may still exhibit features from earlier stages, consistent with explanations. Unlike post-hoc CAM-based methods, which require backpropagation or perturbation for each class, SoftCAM generates class-specific explanations along with prediction in a single forward pass, making it more resource-efficient.

Briefly, both unregularized and sparse models achieved test accuracies comparable to their baseline (Acc $\approx 0.85$ vs. 0.97 on Fundus and OCT), with SoftCAM, particularly the sparse variant, providing the most interpretable class-wise explanations with the best sensitivity. Qualitative explanations for the OCT results are provided in Appendix I.4.

## 5. Discussion

Here, we introduced SoftCAM, a lightweight architectural modification that makes standard black-box CNNs inherently interpretable without relying on post-hoc explanations. SoftCAM replaces the final pooling and fully connected layer with $1\times1$ convolutions, producing class-specific evidence maps that directly inform predictions. This design also supports ElastiNet regularization on the evidence maps to improve explanations. As a result, SoftCAM produces aligned predictions and explanations in a single forward-pass, yielding resource-efficient, self-explainable CNNs. We validated the method on two widely used CNN backbones, ResNet-50 and VGG-16, across three medical imaging modalities—fundus photographs, retinal OCT scans, and Chest X-rays—assessing both classification performance and explainability, and found that the resulting explainable model achieve classification performance comparable to their black-box baselines. For explainability, SoftCAM was benchmarked against six state-of-the-art post-hoc saliency methods—CAM, ScoreCAM, LayerCAM, GradCAM, Guided BP, and Itgd Grad—using qualitative and quantitative metrics that capture both alignment with human knowledge and alignment with the model's true decision-making, demonstrating superior interpretability without compromising predictive performance. In particular, localization precision and the introduced activation sensitivity assess how well explanations match clinically relevant biomarkers using expert annotations, while faithfulness measures how highlighted regions truly influence the model's predictions. Interestingly, our results revealed a discrepancy between human-aligned and model-aligned metrics (Fig. 3): explanations with the highest faithfulness did not always correspond to the strongest alignment with expert annotations. This might suggests two possible explanations: first, the models may rely on clinically relevant signals that are hidden to humans, highlighting the need for evaluation metrics that jointly account for human domain knowledge and the models' internal reasoning; alternatively, occlusion may introduce out-of-distribution inputs that potentially affect sensitivity analysis results, thereby undermining the reliability of the explanations (Hase et al., 2021).

Despite its promising results, SoftCAM has some limitations that warrant further investigation. First, SoftCAM relies on the final low-resolution feature maps (e.g., $16\times16$ for standard CNNs), limiting the spatial granularity needed for fine-grained medical tasks that require lesion- or pixel-level precision. In our case, both ResNet and VGG models use large receptive fields, producing low-resolution feature maps. Consequently, the class-evidence layer generates coarse-grained explanations, though ElasticNet constraints help improve localization. Future work could focus on generating higher-resolution explanations with softCAM to overcome coarseness from downsampling in deep CNNs; integrating Soft-CAM with other architectures like ViTs (Dosovitskiy et al., 2021); extending it to tasks such as weakly supervised segmentations or object detection; and evaluating its utility in other high-stake domains, including agriculture or medical modalities such as dermoscopy for skin cancer detection or MRI for brain tumor detection. Finally, SoftCAM was compared only against a subset of relevant post-hoc methods, consistent with our primary goal of avoiding post-hoc explanations for CNNs classifiers. Future work could extend this evaluation to include self-explainable models, such as part-prototype networks (Chen et al., 2019), concept-based models (Koh et al., 2020), attention-based explainable architectures (Djoumessi et al., 2025), and intrinsically interpretable architectures (Böhle et al., 2022).

## Acknowledgments

This project was supported by the Hertie Foundation, the German Science Foundation (Excellence Cluster EXC 2064 "Machine Learning—New Perspectives for Science", project number 390727645; BE 5601/14-1, project number 571331899). The authors thank the International Max Planck Research School for Intelligent Systems (IMPRS-IS) for supporting Kerol Djoumessi.

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

## Appendix A. Related work

Explainable AI (XAI) methods for image analysis can be categorized into attribution and non-attribution-based approaches (He et al., 2022; Hossain et al., 2023). Attribution-based methods aim to explain "where" important features are located by generating saliency maps that assign importance scores to pixels or regions. In contrast, non-attribution methods focus on explaining "why" a decision was made, typically providing global explanations using approaches such as prototype- or concept-based explanations (Chen et al., 2019; Koh et al., 2020). Attribution-based methods usually provide post-hoc local explanations for black-box models after training. In contrast, non-attribution-based methods are generally inherently interpretable by design (Chen et al., 2019; Brendel and Bethge, 2019; Koh et al., 2020; Böhle et al., 2022), embedding transparency within their architecture. However, inherently interpretable models are typically model-agnostic and often face reduced performance, reflecting a persistent trade-off between interpretability and predictive performance (Rudin, 2019). To bridge this gap, we propose SoftCAM, a protocol that makes traditional black-box CNNs inherently interpretable. Our approach builds on and generalizes prior work.

Aubreville et al. (2019) proposed a dual-branch architecture in which the first branch performed classification using a black-box CNN, while the second branch generated post-hoc explanations, requiring two separate forward passes. In the explanation branch, the global average pooling (GAP) layer is removed and the linear classifier is replaced with convolutional layers that share weights with the main classifier during inference, producing class-specific activation maps. In contrast, SoftCAM integrates both prediction and explanation within a single, end-to-end forward pass, eliminating the need for redundant computation and weights sharing. Similarly, (Donteu et al., 2023) leveraged a convolutional classifier to generate explicit class-evidence maps and applied a sparsity constraint to enhance interpretability in a self-explainable Bag-of-Local-Features model (BagNet) (Brendel and Bethge, 2019). SoftCAM extends and generalizes this concept to standard black-box CNNs, broadening its applicability to various medical imaging tasks and modalities.

## Appendix B. Datasets

We evaluated our approach on three publicly available medical imaging datasets spanning three different modalities: the Kaggle Diabetic Retinopathy (DR) (Dugas et al., 2015), the Retinal OCT dataset (Kermany et al., 2018), and the RSNA Chest X-ray (CXR) dataset (Shih et al., 2019).

- **Kaggle DR Dataset.** This dataset comprises $88,702$ high-resolution retinal fundus images labeled for DR severity on a 5-point scale from 0 (No DR) to 4 (Proliferative DR). After applying an automated quality filtering pipeline using an ensemble of EfficientNet models trained on the ISBI2020[1] challenge dataset, we retained $45,923$ images from $28,984$ subjects. The resulting class distribution was 73%, 15%, 8%, 3%, and 1% for classes 0-4 respectively. For binary classification (early DR detection), we grouped class {0} vs. {1,2,3,4}, yielding an imbalance of 73% vs. 27%. Additionally, lesion annotations for 65 images were obtained from Djoumessi et al. (2024b) for evaluating the model's explanations at localizing DR-related lesions.

---

1. https://isbi.deepdr.org/challenge2.html

- **Retinal OCT Dataset.** This dataset consists of $108,315$ B-scans categorized into four classes: Drusen, Diabetic macular edema (DME), Choroidal neovascularization (CNV), and Normal. A separate test set of $1,000$ B-scans is provided. Following Djoumessi et al. (2024a), we excluded low-resolution scans (width $\leq 496$). As preliminary experiments showed that using the full dataset did not significantly improve performance, we subsampled the training set (by randomly removing half of the healthy images following Djoumessi et al. (2024a)) to $34,962$ scans ($8,616$ Drusen, $26,346$ Normal) for binary classification (Drusen vs. Normal), preserving the original class imbalance ($73\%$ vs $27\%$). Additionally, we used 40 drusen-annotated B-scans from Djoumessi et al. (2024a) to evaluate the model's explanations at localizing drusen lesions. For the multi-class classification task, the training was randomly reduced to $17,200$ images while maintaining the original class distribution: $45\%$ Normal, $34\%$ CNV, $10\%$ DME, and $9\%$ Drusen.

- **RSNA Chest X-ray Dataset.** This dataset includes $30,227$ frontal-view chest radiographs labeled as "Normal", "No Opacity/Not Normal", and "Opacity" (indicative of pneumonia). Pneumonia cases come with bounding box annotations, which facilitate the evaluation of the model's explanations. For our binary classification task, we selected images labeled as either "Normal" or "Opacity", resulting in $14,863$ images with a $60\%$ vs. $40\%$ class distribution.

Each dataset was split into training ($75\%$), validation ($10\%$), and test ($15\%$) sets, except for the Retinal OCT dataset, which followed an $80\%$-$20\%$ training-validation split, due to its predefined test set (250 images per class). All training splits used in our experiments are provided in CSV format and publicly available via the project's GitHub repository[2].

## Appendix C. Implementation details

### C.1. Baseline models

The effectiveness of our method was evaluated using two widely adopted black-box CNN architectures: ResNet-50 and VGG-16. These models were chosen due to their distinct architectures, such as depth, theoretical receptive field size, and classification head design, which allow for a broad assessment of our method's generalizability. In both models, the standard classification head was systematically replaced with our proposed convolution-based evidence map layer to enable inherent interpretability.

For ResNet50, we removed the global average pooling layer and final fully connected layer (FCL), substituting them with a class evidence layer consisting of $C$ convolutional filters ($1 \times 1$, stride 1), where $C$ is the number of output classes. This layer directly produces class-specific evidence maps (Sec. 2.2). For VGG-16, whose classifier head consist of several fully connected layers, each FCL was replaced by an equivalent $1 \times 1$ convolutional layer. Specifically, an FCL of size $b_1 \times b_2$ was transformed into a convolutional layer of size $b_1 \times b_2 \times 1 \times 1$, preserving the original parameter count and model capacity. These architectural changes maintain model complexity and capacity while introducing interpretability directly into the classification mechanism.

## C.2. Data preprocessing and augmentation

Fundus images were preprocessed by cropping them to a square shape using a circle-fitting method as described in (Mueller et al.). All datasets were then resized to $512 \times 512$ pixels, except for the retinal OCT dataset, which was resized to $496 \times 496$ to better match its original lower resolution. Image intensities were normalized using the mean and standard deviation computed from the respective training sets.

During training, standard transformations were applied across all datasets. These included flipping, rotation, random cropping, and translation, each applied with a fixed probability. For the Kaggle dataset, which contains color fundus images, additional color augmentations were introduced to improve generalization.

## C.3. Training setup

All models were sourced from Torchvision and initialized with pretrained weights from ImageNet. They were subsequently fine-tuned on each dataset using a consistent training setup[2]. Following Djoumessi et al. (2024a) and Donteu et al. (2023), we employed the cross-entropy loss function and optimized model parameters using stochastic gradient descent with Nesterov momentum (momentum factor of 0.9). The initial learning rate was set to 1.10-3, and a clipped cosine annealing learning rate scheduling was applied with the minimum learning rate set to $1.10^{-4}$. Weight decay was set to $5.10^{-4}$. The training was conducted for 70 epochs with a mini-batch size of 16 on an NVIDIA A40 GPU using PyTorch.

## Appendix D. Baseline post-hoc methods

SoftCAM was benchmarked against widely used post-hoc methods, spanning CAM-based and backpropagation-based approaches. CAM-based methods produce coarse-grained explanations due to their reliance on low-resolution convolutional feature maps, whereas backpropagation-based techniques provide fine-grained pixel-level attributions that perverse full spatial resolutions. CAM-based methods include gradient-based approaches such as GradCAM and LayerCAM, as well as the gradient-free method ScoreCAM. Backpropagation-based techniques include Integrated Gradients and Guided Backpropagation. Gradient-based methods primarily differ in how they aggregate gradients to compute importance weights, while gradient-free methods vary in how these weights are estimated without backpropagation.

Guided Backpropagation and Integrated Gradients have consistently performed well in generating saliency maps to explain black-box CNN classifiers on retinal images (Ayhan et al., 2022; Djoumessi et al., 2024b), while GradCAM has shown strong localization performance for chest X-ray interpretation (Saporta et al., 2022). Below is a brief description of the five post-hoc methods used.

**ScoreCAM** (Wang et al., 2020). A gradient-free method that eliminates the need for gradient information by assessing the importance of each activation map based on its forward-pass contribution to the target class score, and produces the final output via a weighted sum of these maps.

---

2. The code is available at https://github.com/kdjoumessi/SoftCAM

**LayerCAM** (Jiang et al., 2021). A gradient-based method that generates class activation maps by leveraging the element-wise product of ReLU-activated gradients and feature maps at any convolutional layer, enabling fine-grained, spatially precise visual explanations without requiring global average pooling.

**GradCAM** (Selvaraju et al., 2017). A gradient-based approach that uses the gradients of the target class flowing into the final convolutional layer to produce a coarse localization map, highlighting important regions in the image by upsampling the resulting map.

**Guided backpropagation (Guided BP)** (Springenberg et al., 2014). A gradient-based approach that modifies the standard backpropagation process to propagate only positive gradients through positive activations, producing fine-grained visualizations that highlight features strongly activating specific neurons in relation to the target output.

**Integrated Gradient (Itgt Grad.)** (Sundararajan et al., 2017). A gradient-based method that attributes model predictions to input features by computing the path integral of gradients along a straight-line path from a baseline to the actual input, yielding fine-grained explanations.

ScoreCAM and LayerCAM were implemented with TorchCAM (Fernandez, 2020), while the other methods were implemented from Captum (Kokhlikyan et al., 2020).

## Appendix E. Explainability metrics

For binary tasks, performance was evaluated using accuracy and AUC, while for multi-class tasks, accuracy and the quadratic Cohen's kappa score ($\kappa$) were used. The AUC measures class separability, whereas the kappa score captures the agreement beyond chance.

Explainability was assessed using several quantitative metrics, including activation consistency (Donteu et al., 2023), top-k localization precision (Donteu et al., 2023), activation precision (Barnett et al., 2021), further extended to activation sensitivity, and faithfulness (Yeh et al., 2019). Together, these metrics quantify both alignment with expert clinical knowledge (top-k localization, activation precision, and activation sensitivity) and alignment with the model's true decision-making process (faithfulness).

### E.1. Activation consistency

The activation consistency (Donteu et al., 2023) measures how well local explanations (i.e., positive or negative activation in attribution maps) reflect the true disease or healthy labels across a dataset. Intuitively, an interpretable model should consistently show positive activation regions associated with pathology for disease samples, while producing minimal or negative activations for healthy samples. This metric therefore assesses whether explanation maps globally align with the semantic meaning implied by ground-truth labels.

Following Donteu et al. (2023), activation consistency is computed as the proportion of positive activations in the attribution maps of disease samples and the proportion of negative activations in the attribution maps of healthy samples, averaged over the test set. A higher score indicates more coherent and label-aligned explanations.

Formally, let $\mathcal{D} = \mathcal{D}_{\text{disease}} \cup \mathcal{D}_{\text{healthy}}$ denote the test set, and let be $A_i(x, y)$ be attribution map sample $i$. Given the following indicator functions

$$\mathbb{1}_+(A_i) = \frac{1}{|A_i|} \sum_{(x,y)} \mathbb{1}\big(A_i(x, y) > 0\big), \quad \mathbb{1}_-(A_i) = \frac{1}{|A_i|} \sum_{(x,y)} \mathbb{1}\big(A_i(x, y) < 0\big),$$

corresponding to the proportion of positive and negative activations in the attribution map. **Activation consistency** is then defined as

$$AC_+ = \frac{1}{|\mathcal{D}_{\text{disease}}|} \sum_{i \in \mathcal{D}_{\text{disease}}} \mathbb{1}_+(A_i); \tag{5}$$

$$AC_- = \frac{1}{|\mathcal{D}_{\text{healthy}}|} \sum_{i \in \mathcal{D}_{\text{healthy}}} \mathbb{1}_-(A_i), \tag{6}$$

where $AC_+$ measures how consistently explanations highlight pathological regions in disease samples, $AC_-$ measures how consistently they suppress activations in healthy samples. This provides a dataset-level assessment of whether local explanations globally support the model's classification behavior.

### E.2. Top-k localization precison

Top-k localization precision (Donteu et al., 2023) measures how well an explanation map highlights clinically relevant regions that overlap with ground-truth annotations. Specifically, it computes the proportion of the top-k positively activated regions that coincide with annotated areas obtained from clinicians.

Given an explanation map, we first upsampled it to the original input resolution and split into non-overlapping patches of size $33 \times 33$. Each patches is assigned a saliency score equal to the average activation within that patch. The top-k ($k \in [1, 30]$) most salient patches are then selected, and the metric computes the fraction of these patches that overlap with annotated ground-truth regions. This generalizes the "pointing game" metric (Zhang et al., 2018), which only considers the single most salient region (top-1), making it better suited for medical imaging tasks where disease-relevant features (e.g., retinal lesions or other pathological markers) are often spatially distributed across the image.

Formally, let $S$ denote the upsampled explanation map and let the image be partitioned into $P$ non-overlapping patches $\{P_1, \ldots, P_p\}$. The average activation of each patch is

$$a_p = \frac{1}{|P_p|} \sum_{(x,y) \in P_p} S(x, y).$$

Define the indices of the top-k salient patches as $T_k = \arg \text{top}_k \{a_1, \ldots, a_p\}$. Let $G$ be the binary ground-truth annotation mask. Patch-annotation is defined by

$$\mathbb{1}_{\text{overlap}}(P_p, G) = \begin{cases} 1, & \text{if } \sum_{(x,y) \in P_p} G(x, y) > 0, \\ 0, & \text{otherwise.} \end{cases}$$

The **Top-k localization precision** is then defined as

$$\text{Top-k precision}(k) = \frac{1}{k} \sum_{p \in T_k} \mathbb{1}_{\text{overlap}}(P_p, G). \tag{7}$$

A higher score indicates that the explanation consistently highlights clinically meaningful regions identified by experts.

### E.3. Activation precision and activation sensitivity

Let $\mathcal{X} = \{\mathbf{X}\}_{i=1}^{n}$ denote a set of input images, $\mathcal{M} = \{\mathbf{M}\}_{i=1}^{n}$ their corresponding binary segmentation masks, and $\mathcal{S} = \{\mathbf{S}\}_{i=1}^{n}$ the explanation or saliency maps produced by a given method. *Activation precision* measures how much of the positive evidence highlighted by an explanation map falls within the annotated ground-truth region (Barnett et al., 2021).

Before computing the metric, saliency maps are preprocessed by thresholding negative values to zero, $\mathbf{S}_i^+ = \max(\mathbf{S}_i, 0)$, ensuring that only positive evidence contributes to the evaluation. For each sample, activation precision (AP) is defined as the proportion of positive activation mass contained inside the annotation mask:

$$\mathbf{AP}_i = \frac{\sum_p \mathbf{S}_i^+(p)\mathbf{M}_i(p)}{\sum_p \mathbf{S}_i^+(p) + \epsilon},$$

where $p$ indexes spatial locations and $\epsilon$ is a small constant preventing division by zero. The dataset-level activation precision is then obtained by averaging over all samples:

$$\mathbf{AP} = \frac{1}{n}\sum_{i=1}^{n}\mathbf{AP}_i. \tag{8}$$

This metric captures how precisely the explanations signal aligns with expert annotation, providing a clinically meaningful measure of the quality of an explanation method.

However, activation precision does not penalize false negatives (i.e. missed relevant regions). To address this, we introduce *activation sensitivity*, which captures the completeness of the explanation by computing the fraction of annotated regions that are covered by the saliency map. We defined per-sample activation sensitivity (AS) by computing the proportion of mask pixels that receive any positive activation:

$$\mathbf{AS}_i = \frac{\sum_p \mathbf{S}_i^+(p)\mathbf{M}_i(p)}{\sum_p \mathbf{M}_i(p) + \epsilon},$$

The dataset-level activation sensitivity is obtained by averaging over samples:

$$\mathbf{AS} = \frac{1}{n}\sum_{i=1}^{n}\mathbf{AS}_i. \tag{9}$$

Intuitively, $\mathbf{AS}_i$ is the fraction of the annotated region that the explanation covers with higher values meaning fewer missed regions.

Unlike activation precision, activation sensitivity penalizes weak activations inside the annotated region. For example, if $M_i(p) = 1$ and $0 < S_i(p) < 1$, the low activation contributes only minimally to the numerator, indicating reduced confidence in that clinically relevant area. This makes activation sensitivity particularly valuable in settings where complete lesion coverage is essential.

### E.4. Failthfulness

Faithfulness, also referred to as sensitivity or fidelity (Yeh et al., 2019), evaluates how well an explanation reflects the model's true decision-making process. It assesses whether the importance assigned to input features corresponds to their actual influence on the model's prediction.

In our implementation, we evaluated faithfulness on correctly classified test samples. Each explanation map is upsampled to the input resolution and split into non-overlapping $33 \times 33$ patches, which are ranked by mean activation values. The top-k most salient patches are iteratively occluded, and after each step we record the relative drop in the model's confidence for the predicted class. This produces a deletion curve from which we computed the Area Under the Deletion Curve (AUDC). A lower AUDC indicates a more faithful explanation, as it reflects a sharper decrease in confidence when the regions deemed important are removed, showing that these regions effectively contributed to the model's decision.

## Appendix F. Sparsity regularization selection for the binary tasks

The sparsity regularization coefficient $\lambda_1$ in Eq. 4 controls the sparsity of the class evidence map, encouraging the model to localize disease regions with high precision. For each task, $\lambda_1$ was selected based on a trade-off between accuracy and AUC on the corresponding validation set, choosing the highest values for which classification performance did not degrade significantly (Fig. 11).

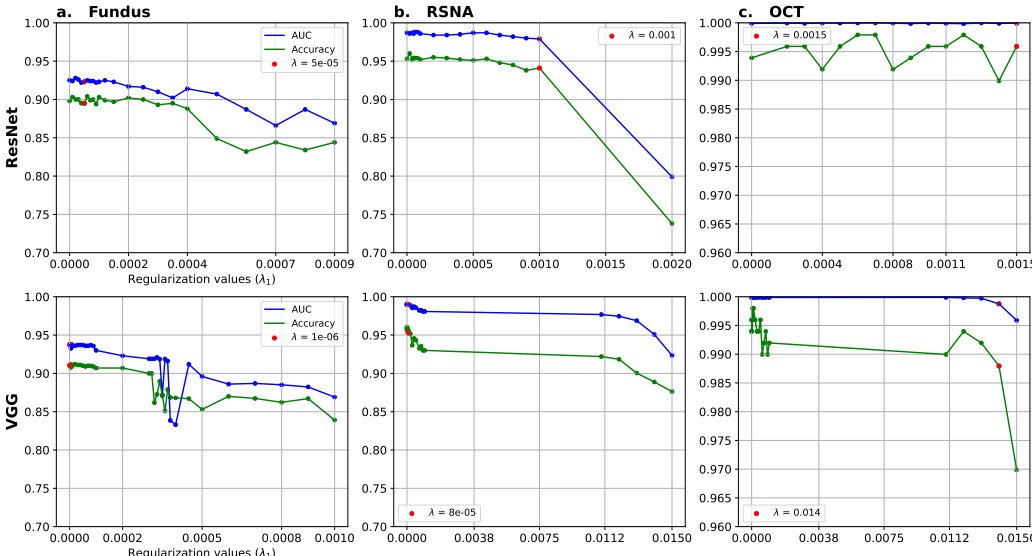

Figure 7: **Model selection on validation sets under varying sparsity regularization strengths.** The regularization coefficient $\lambda$ influences model performance, with notable effects on some datasets but minimal impact on the OCT dataset. The red markers indicate the selected $\lambda$ values, chosen to balance sparsity and classification performance.

## Appendix G. Additionnal Results

**Visualizing explanations.** The evidence map generated by SoftCAM is upsampled to the input resolution for visualization. Like most CAM-based methods, such as GradCAM, ScoreCAM, and LayerCAM that operate on the final convolutional layer, SoftCAM's explanations are limited by the resolution of the backbone (e.g., 16×16 for VGG-16/ResNet-50 with 512×512 input) due to pooling and striding, leading to lower-resolution saliency maps. However, by introducing the class evidence and classification layer directly on top of the low-resolution features and applying regularization, SoftCAM improves spatial precision. In contrast, gradient-based methods like Integrated Gradients (Sundararajan et al., 2017) and Guided Backpropagation (Springenberg et al., 2014) produce high-resolution saliency maps by computing pixel-level gradients, which may lead to noisy maps, especially when the region of interest spans a broader area, as commonly observed in Chest X-ray images.

**Comparison with other approaches.** Unlike post-hoc attribution-based approaches, SoftCAM is inherently interpretable from the classification layer and maintains performance comparable to its black-box counterpart, without a significant trade-off, even when regularization is applied to enhance explainability. Compared to Donteu et al. (2023), SoftCAM extends from interpretable bag-of-local models to general black-box CNN architectures and generalizes the regularization from lasso to ElasticNet, with extensive evaluations across multiple datasets using a broad range of explanability metrics. Compared to Aubreville et al. (2019), our method is trained end-to-end and does not require post-hoc processing, weight sharing between branches, or an additional forward pass to generate explanations.

### G.1. SoftCAM provides inherently interpretable visual explanations

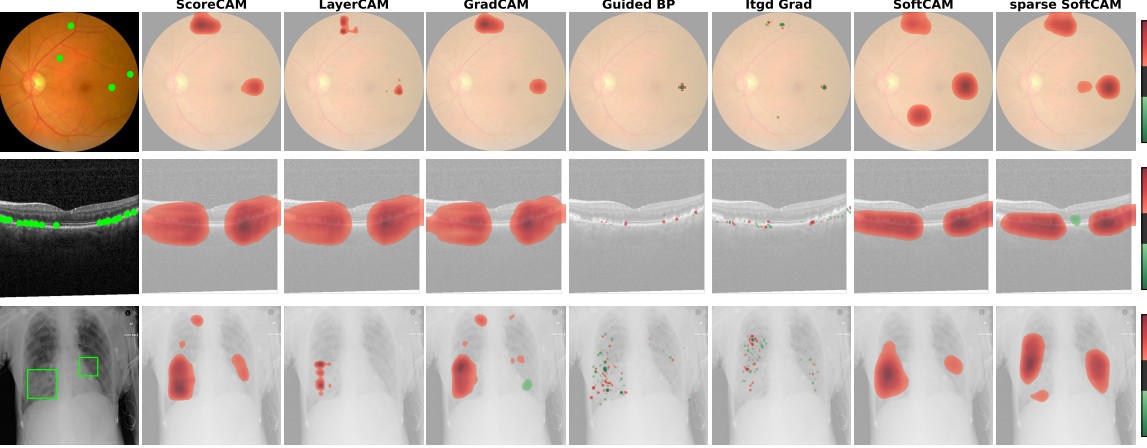

Figure 8: **Explanations generated by different methods from VGG-16**. The first column shows disease images with reference annotations, indicated by green markers or bounding boxes. Each row, from top to bottom, corresponds to fundus, OCT, and Chest X-ray images, respectively. The next five columns present saliency maps generated by post-hoc explanation methods. The final two columns showcase our proposed inherently interpretable SoftCAM explanations.

ADDITIONAL QUALITATIVE EXAMPLES

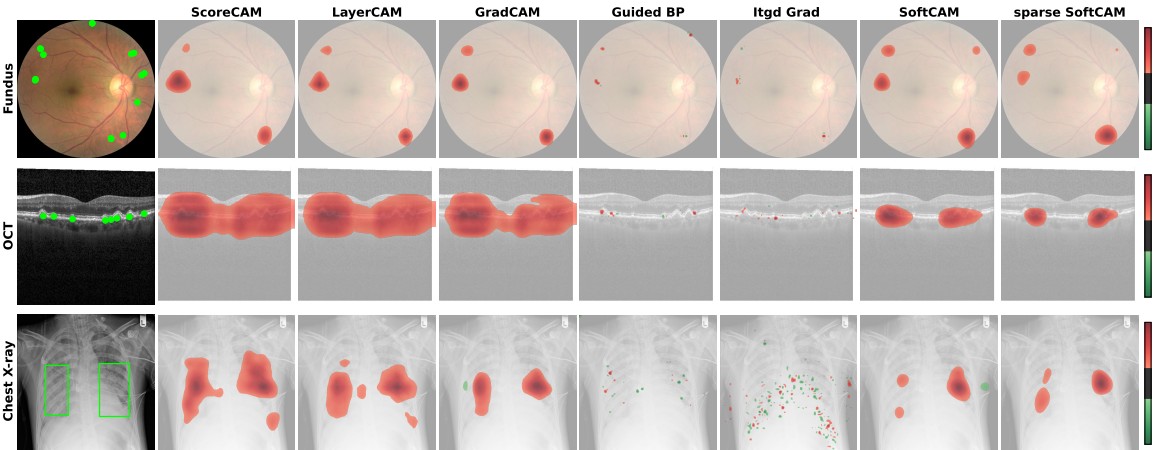

Figure 9: **Additional qualitative explanations from the ResNet model**.

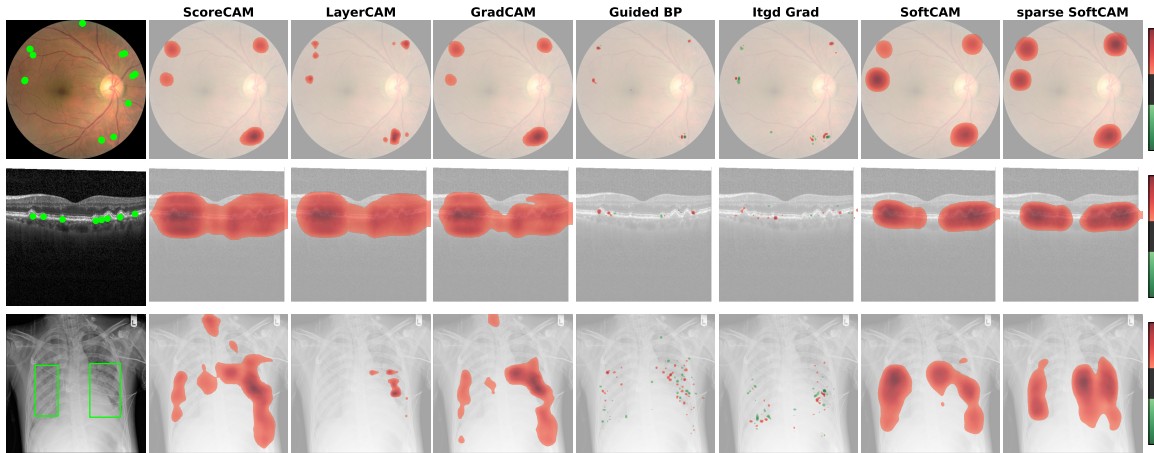

Figure 10: **Additional qualitative explanations from the VGG model**.

### G.2. Activation consistency

We quantified activation consistency only for the SoftCAM variants, as post-hoc methods are not inherently explainable, meaning their explanations do not directly influence the model's decision-making process.

The results align well with qualitative visualizations. On the Fundus dataset, the sparse SoftCAM model exhibits a higher proportion of positive activations with the ResNet backbone, attributed to reduced false positives from the dense model, and fewer negative activations, reflecting the suppression of low-importance activations to zero. On the VGG backbone, regularization primarily reduces false-positive activations from the unregularized

model but leads to a slight increase in activations on healthy samples. A similar result can be observed on the RSNA dataset.

On the OCT dataset, the unregularized SoftCAM with the ResNet backbone generally produces coarse-grained evidence around lesion areas. In contrast, the sparse variant refines these explanations, resulting in lower positive and negative activations across both disease and healthy samples, suggesting more selective and focused localization. However, with the VGG backbone, a higher proportion of negative activations is observed, reflecting the impact of the regularization strength—highlighting the importance of appropriately tuning this parameter for different architectures.

Table 2: Activation consistency on the ResNet model. $\mathbf{r_{LG}^+}$ denotes the proportion of positive or disease activations from disease images, while $\mathbf{r_{LG}^-}$ refers to the proportion of negative or healthy activations from healthy images.

|  | Fundus | | OCT | | RSNA | |
| --- | --- | --- | --- | --- | --- | --- |
|  | $\mathbf{r_{LG}^+}\uparrow$ | $\mathbf{r_{LG}^-}\uparrow$ | $\mathbf{r_{LG}^+}\uparrow$ | $\mathbf{r_{LG}^-}\uparrow$ | $\mathbf{r_{LG}^+}\uparrow$ | $\mathbf{r_{LG}^-}\uparrow$ |
| SoftCAM | $0.28 \pm 0.1$ | $0.86 \pm 0.1$ | $0.30 \pm 0.1$ | $0.85 \pm 0.1$ | $0.75 \pm 0.1$ | $0.47 \pm 0.1$ |
| sparse SoftCAM | $0.55 \pm 0.2$ | $0.76 \pm 0.2$ | $0.23 \pm 0.1$ | $0.83 \pm 0.1$ | $0.79 \pm 0.1$ | $0.45 \pm 0.1$ |

Table 3: Activation consistency on the VGG model. $\mathbf{r_{LG}^+}$ denotes the proportion of positive or disease activations from disease images, while $\mathbf{r_{LG}^-}$ refers to the proportion of negative or healthy activations from healthy images.

|  | Fundus | | OCT | | RSNA | |
| --- | --- | --- | --- | --- | --- | --- |
|  | $\mathbf{r_{LG}^+}\uparrow$ | $\mathbf{r_{LG}^-}\uparrow$ | $\mathbf{r_{LG}^+}\uparrow$ | $\mathbf{r_{LG}^-}\uparrow$ | $\mathbf{r_{LG}^+}\uparrow$ | $\mathbf{r_{LG}^-}\uparrow$ |
| SoftCAM | $0.32 \pm 0.2$ | $0.93 \pm 0.1$ | $0.75 \pm 0.11$ | $0.51 \pm 0.1$ | $0.75 \pm 0.1$ | $0.51 \pm 0.1$ |
| sparse SoftCAM | $0.28 \pm 0.2$ | $0.94 \pm 0.1$ | $0.35 \pm 0.14$ | $0.95 \pm 0.1$ | $0.35 \pm 0.1$ | $0.95 \pm 0.1$ |

Overall, the effect of regularization on the explanations varies depending on the backbone architecture. Nevertheless, the activation consistency metric aligns well with the qualitative explanations, generally capturing the impact of regularization across the dataset for a given architecture.

### G.3. Precision and sensitivity analysis

We quantitatively evaluate the explanations generated by various methods using the ResNet and VGG backbones on the RSNA dataset. With the ResNet model, the unregularized SoftCAM achieves the highest localization precision, whereas the sparse SoftCAM yields the best results in terms of sensitivity. This discrepancy underscores the importance of developing evaluation metrics that balance human-aligned localization quality with model fidelity, capturing both interpretability and decision relevance.

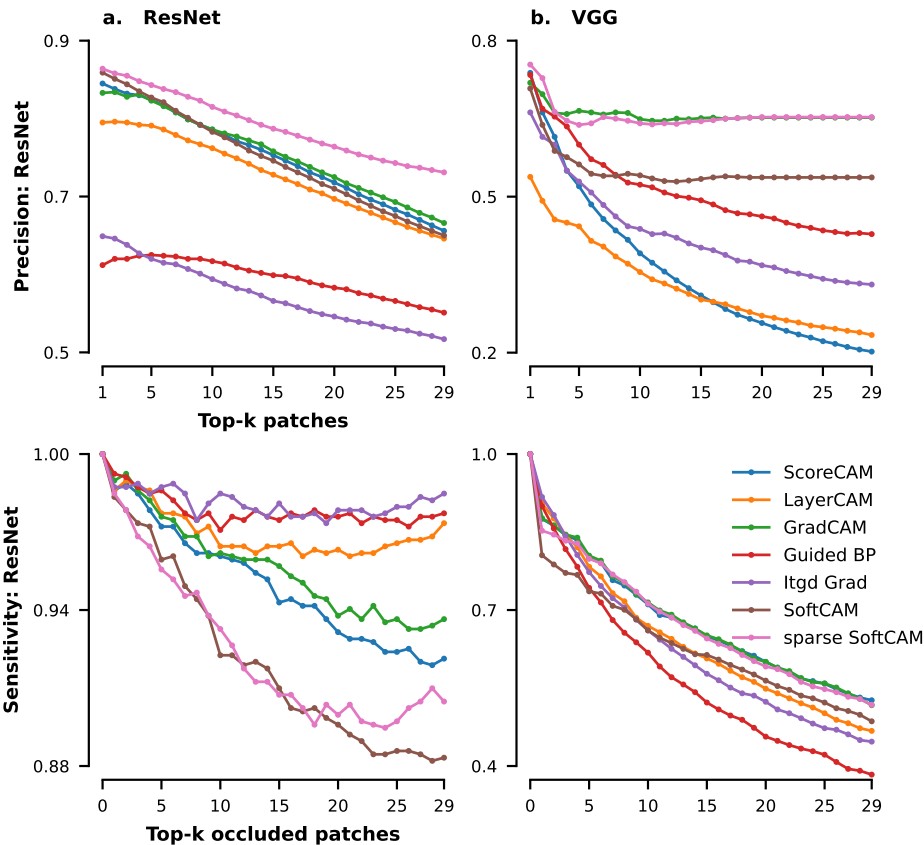

Figure 11: **Precision vs. sensitivity analysis on the RSNA dataset**. Quantitative evaluation of explanations generated by different methods from the ResNet and VGG models on the RSNA dataset.

### G.4. SoftCAM provides localized and faithful explanations

Alongside the visualization comparing SoftCAM variants with post-hoc methods, we provide the corresponding quantitative metrics, reporting both top-k precision and faithfulness.

Table 4: Top-15 localization precision and sensitivity. Sensitivity is quantified as the Area Under the Deleted Curve (AUDC), where lower values indicate greater faithfulness. For precision, higher values indicate better alignment between saliency maps and ground truth annotations. We refer to AUDC as "Del" and Top-K as "Top".

| | ResNet (Top ↑, Del ↓) | | | | | | VGG (Top ↑, Del ↓) | | | | | |
|---|---|---|---|---|---|---|---|---|---|---|---|---|
| | Fundus | | OCT | | RSNA | | Fundus | | OCT | | RSNA | |
| | Top | Del | Top | Del | Top | Del | Top | Del | Top | Del | Top | Del |
| ScoreCAM | 0.22 | 0.77 | 0.12 | 0.84 | 0.72 | 0.98 | 0.30 | 0.78 | 0.12 | 0.76 | **0.74** | 0.93 |
| LayerCAM | 0.25 | 0.76 | 0.11 | 0.85 | 0.75 | 0.97 | 0.30 | 0.76 | 0.11 | 0.76 | 0.72 | 0.91 |
| GradCAM | 0.37 | 0.75 | 0.15 | 0.84 | 0.75 | 0.97 | **0.65** | 0.79 | 0.15 | 0.77 | 0.72 | 0.92 |
| Guided BP | 0.38 | **0.69** | 0.36 | 0.80 | 0.60 | 0.98 | 0.48 | **0.72** | 0.36 | **0.65** | 0.66 | 0.92 |
| Itgd Grad | 0.34 | 0.73 | 0.30 | 0.82 | 0.56 | 0.98 | 0.40 | 0.75 | 0.30 | 0.77 | 0.61 | 0.93 |
| SoftCAM | 0.40 | 0.79 | 0.46 | 0.84 | 0.74 | **0.95** | 0.54 | 0.73 | 0.72 | 0.68 | **0.74** | 0.92 |
| sparse SoftCAM | **0.52** | 0.78 | **0.86** | **0.57** | **0.78** | **0.95** | **0.65** | 0.78 | **0.82** | 0.66 | 0.71 | **0.90** |

Table 5: Top-k localization precision and sensitivity, $k = 30$. Sensitivity is quantified as the Area Under the Deleted Curve (AUDC), where lower values indicate greater faithfulness—that is, a larger drop in the model's confidence when the most relevant patches are removed. For precision, higher values indicate better alignment between saliency maps and ground truth annotations.

| | ResNet (Top ↑, Del ↓) | | | | | | VGG (Top ↑, Del ↓) | | | | | |
| | Fundus | | OCT | | RSNA | | Fundus | | OCT | | RSNA | |
| | Top | Del | Top | Del | Top | Del | Top | Del | Top | Del | Top | Del |
| ScoreCAM | 0.16 | 0.67 | 0.07 | 0.73 | 0.66 | 0.97 | 0.20 | 0.67 | 0.08 | 0.55 | 0.62 | 0.88 |
| LayerCAM | 0.22 | 0.65 | 0.08 | 0.74 | 0.65 | 0.97 | 0.23 | 0.64 | 0.08 | 0.56 | **0.65** | 0.84 |
| GradCAM | 0.37 | 0.64 | 0.14 | 0.73 | 0.67 | 0.95 | **0.65** | 0.68 | 0.14 | 0.58 | 0.61 | 0.86 |
| Guided BP | 0.30 | **0.57** | 0.23 | 0.68 | 0.55 | 0.97 | 0.43 | 0.57 | 0.23 | **0.40** | 0.58 | 0.85 |
| Itgd Grad | 0.28 | 0.63 | 0.20 | 0.70 | 0.52 | 0.98 | 0.33 | **0.62** | 0.2 | 0.51 | 0.55 | 0.88 |
| SoftCAM | 0.39 | 0.69 | 0.46 | 0.61 | 0.65 | **0.92** | 0.54 | 0.63 | 0.72 | 0.45 | 0.64 | 0.84 |
| sparse SoftCAM | **0.52** | 0.68 | **0.86** | **0.31** | **0.73** | 0.93 | **0.65** | 0.674 | **0.82** | 0.43 | 0.63 | **0.82** |

# Appendix H. Activation precision and sensitivity on the RSNA dataset

## H.1. Lasso vs Ridge penalty

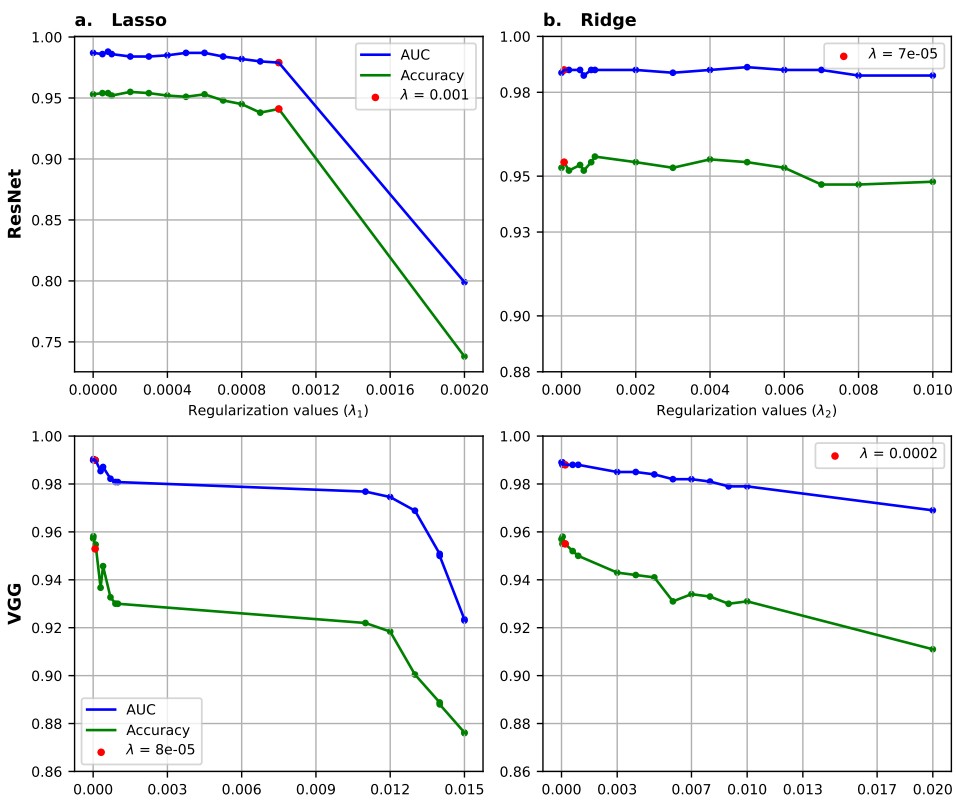

Figure 12: **Model selection on validation sets under varying values.** The regularization strengths $\lambda_1$ and $\lambda_2$ influence model performance. The red markers indicate the selected regularization values, chosen to balance classification performance.

### H.2. Activation precision vs. activation sensitivity

Table 6: Activation Precision (AP) vs. Activation Sensitivity (AS) for different SoftCAM variants and baseline post-hoc methods. The base SoftCAM consistently lies between the lasso and ridge variants, highlighting the importance of balancing $\ell_1$ and $\ell_2$ values to achieve an optimal trade-off between precision and completeness.

|                | ResNet   |          | VGG      |          |
|----------------|----------|----------|----------|----------|
|                | AP ↑     | AS ↑     | AP ↑     | AS ↑     |
| ScoreCAM       | 0.470    | 0.318    | 0.403    | 0.303    |
| LayerCAM       | 0.456    | 0.300    | 0.401    | 0.120    |
| GradCAM        | 0.525    | 0.252    | 0.373    | 0.260    |
| Guided BP      | 0.381    | 0.033    | 0.364    | 0.044    |
| Itgd Grad.     | 0.286    | 0.040    | 0.322    | 0.039    |
| SoftCAM        | 0.526    | 0.251    | 0.461    | 0.355    |
| ridge SoftCAM  | 0.440    | **0.316**| 0.412    | **0.396**|
| sparse SoftCAM | **0.654**| 0.182    | **0.519**| 0.320    |

### H.3. More examples: activation precision vs activation sensitivity

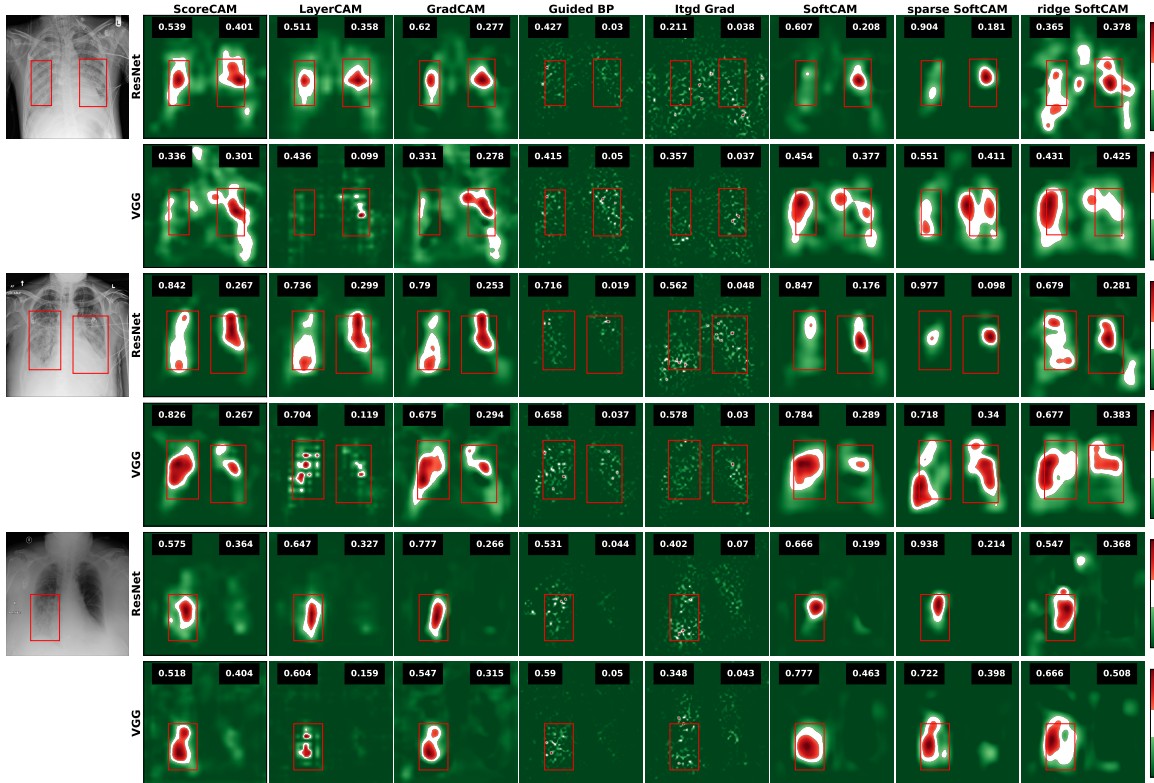

Figure 13: **Localization precision and sensitivity for pneumonia detection**. Each column shows explanations generated by different methods. Ground-truth bounding boxes are drawn on each map, with the top-right value indicating the activation precision, while the top-left value indicates the activation sensitivity. This shows the trade-off between ridge and lasso regularization.

## Appendix I. Multi-class analysis

We extended our method to the multi-class setting for retinal disease diagnosis, using the same training setup as in the binary tasks. The only modification was adjusting the number of output classes in the evidence layer to 5 for DR grading (fundus dataset) and 4 for retinal disease classification (OCT dataset, and selecting appropriate lasso penalties (e.g. $\lambda_1 = 9.10^{-4}$ vs. $\lambda_1 = 3.10^{-6}$ for ResNet and VGG on the OCT dataset).

### I.1. Regularization

Given the small size of retinal lesions, we used Lasso regularization, selecting $\lambda_1$ values that balanced performance (Fig. 14)

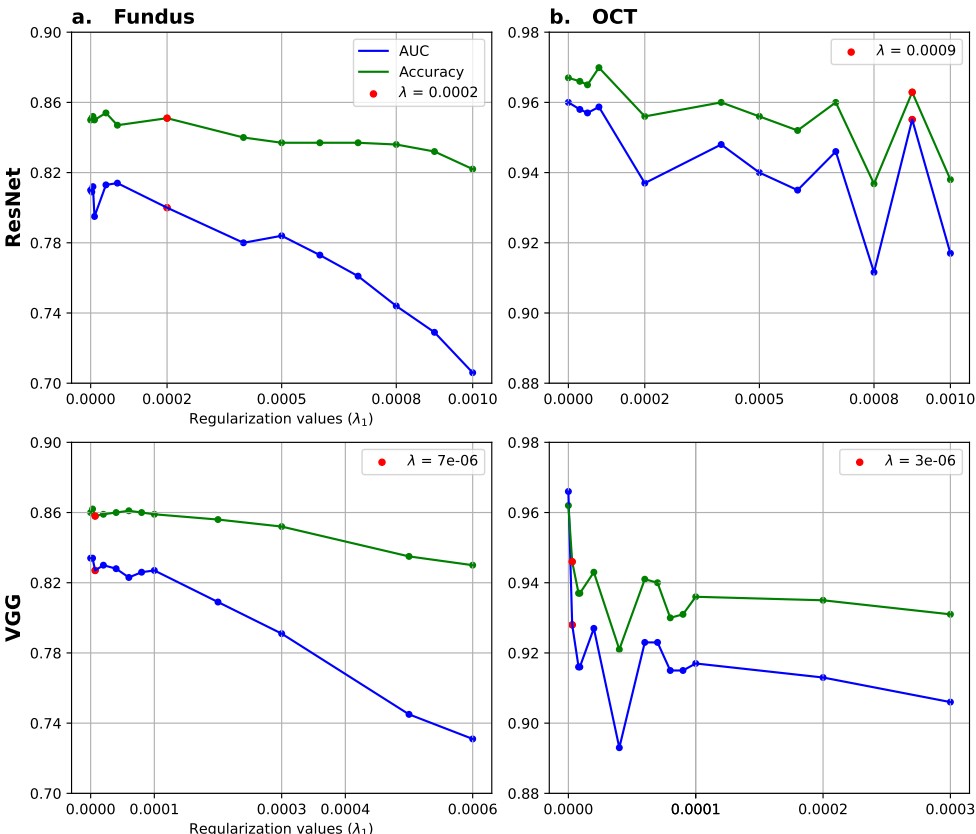

Figure 14: **Model selection on validation sets under varying sparsity values.** The regularization coefficients $\lambda_1$ influence model performance. The red markers indicate the selected regularization values to balance classification performance.

### I.2. Area Under the Deleted Curve

The relative area under the deletion curve (AUDC) was computed from the sensitivity analysis by occluding the top-30 patches ranked by importance in the explanation map

(Fig. 5). In both tasks, the dense and sparse SoftCAM achieved superior performance, with sparse SoftCAM yielding the lowest AUDC, indicating the highest faithfulness (Tab. 7).

Table 7: Area Under the Deleted Curve (AUDC ↓).

|  | ResNet | | VGG | |
| --- | --- | --- | --- | --- |
|  | Fundus | OCT | Fundus | OCT |
| ScoreCAM | 0.894 | 0.819 | 0.880 | 0.852 |
| LayerCAM | 0.889 | 0.817 | **0.869** | 0.850 |
| GradCAM | 0.887 | 0.815 | 0.872 | 0.847 |
| Guided BP | 0.899 | 0.793 | 0.905 | 0.823 |
| Itgd Grad | 0.907 | 0.821 | 0.901 | 0.833 |
| dense SoftCAM | 0.870 | 0.825 | 0.905 | 0.826 |
| sparse SoftCAM | **0.856** | **0.609** | 0.882 | **0.806** |

### I.3. Qualitative explanation on retinal fundus images

For the multi-class DR detection tasks on fundus images, SoftCAM variants produced more focused and class-consistent explanations. Alongside the sparse and unregularized evidence maps, we also include visualizations from post-hoc methods.

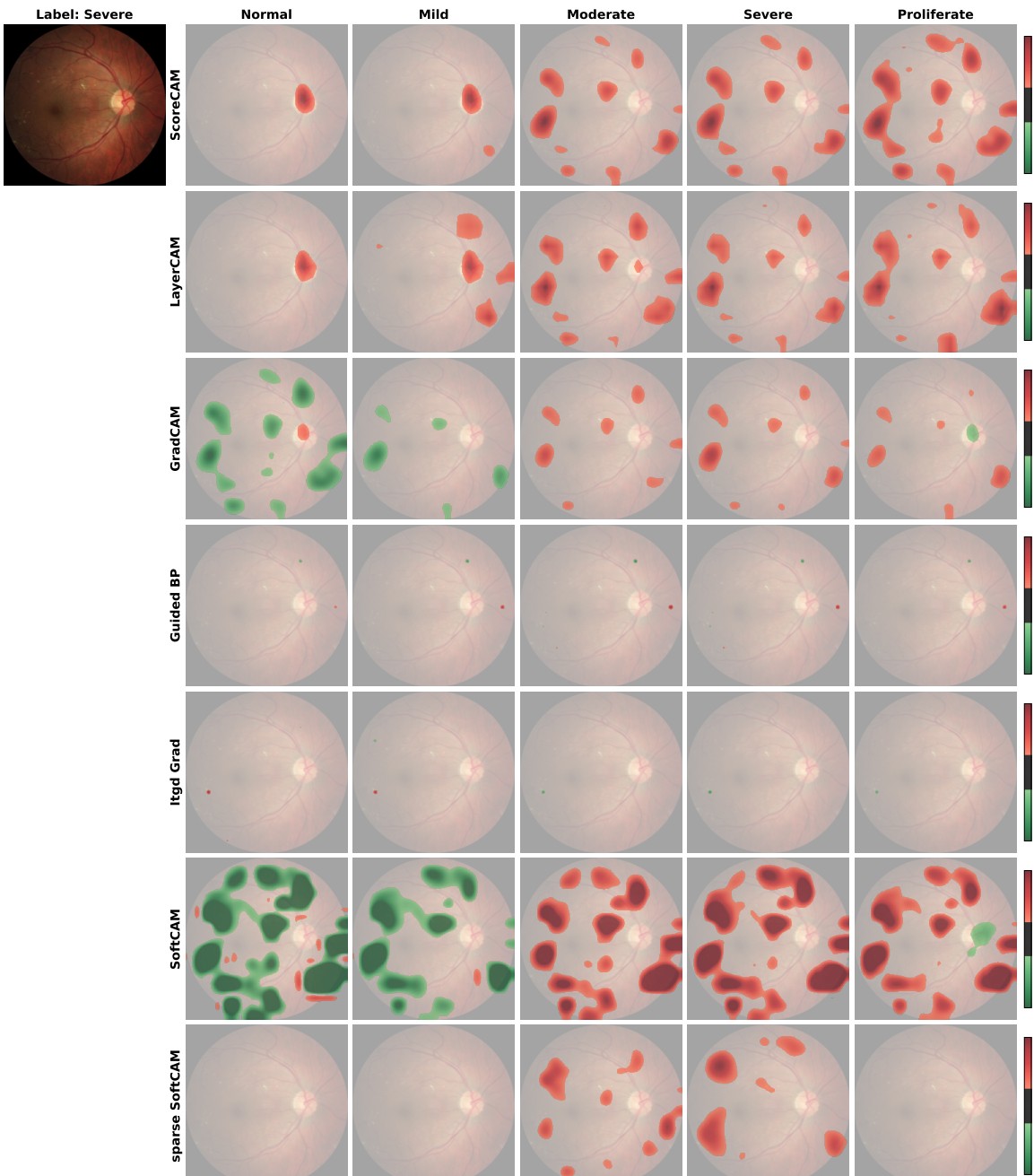

Figure 15: **Class-specific explanation with the ResNet backbone**. The application of our method to multi-class DR detection demonstrates the utility of class-specific explanations produced by the sparse SoftCAM, which more precisely highlight disease-relevant regions compared to the dense SoftCAM and the best-performing post-hoc method, GradCAM. In the example shown, the image is labeled as severe DR, and the highlighted regions correspond to suspicious areas, reflecting relevant DR lesions.

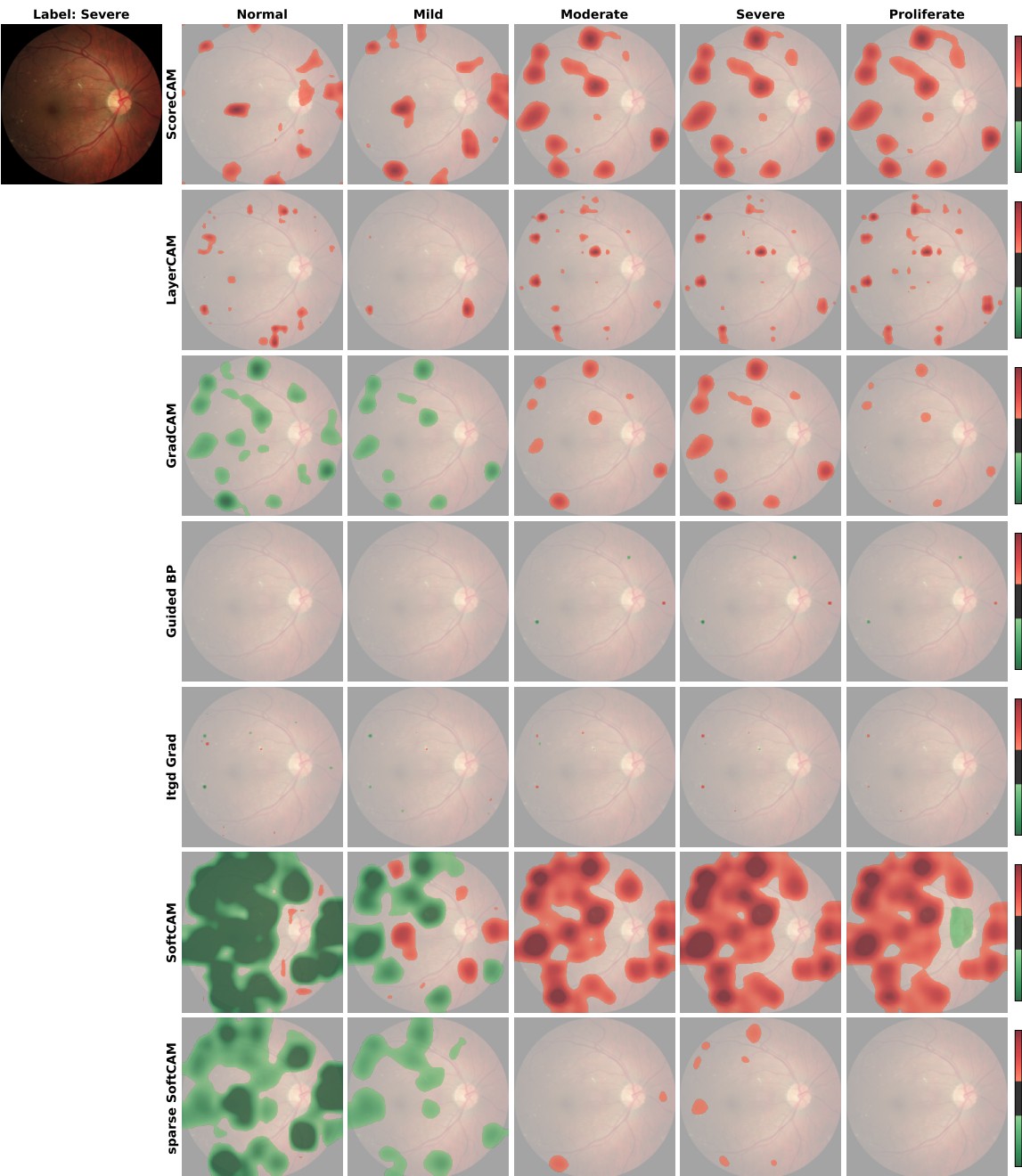

Figure 16: **Class-specific explanation with the VGG backbone**.

## I.4. Qualitative explanation on retinal OCT images

For multi-class OCT-based retinal disease classification, SoftCAM variants yielded more focused and class-consistent explanations. We also provide visualizations from other methods. Note that GradCAM and Guided BP were the best-performing post-hoc methods for ResNet and VGG according to the Area Under the Deletion Curve (Tab. 7).

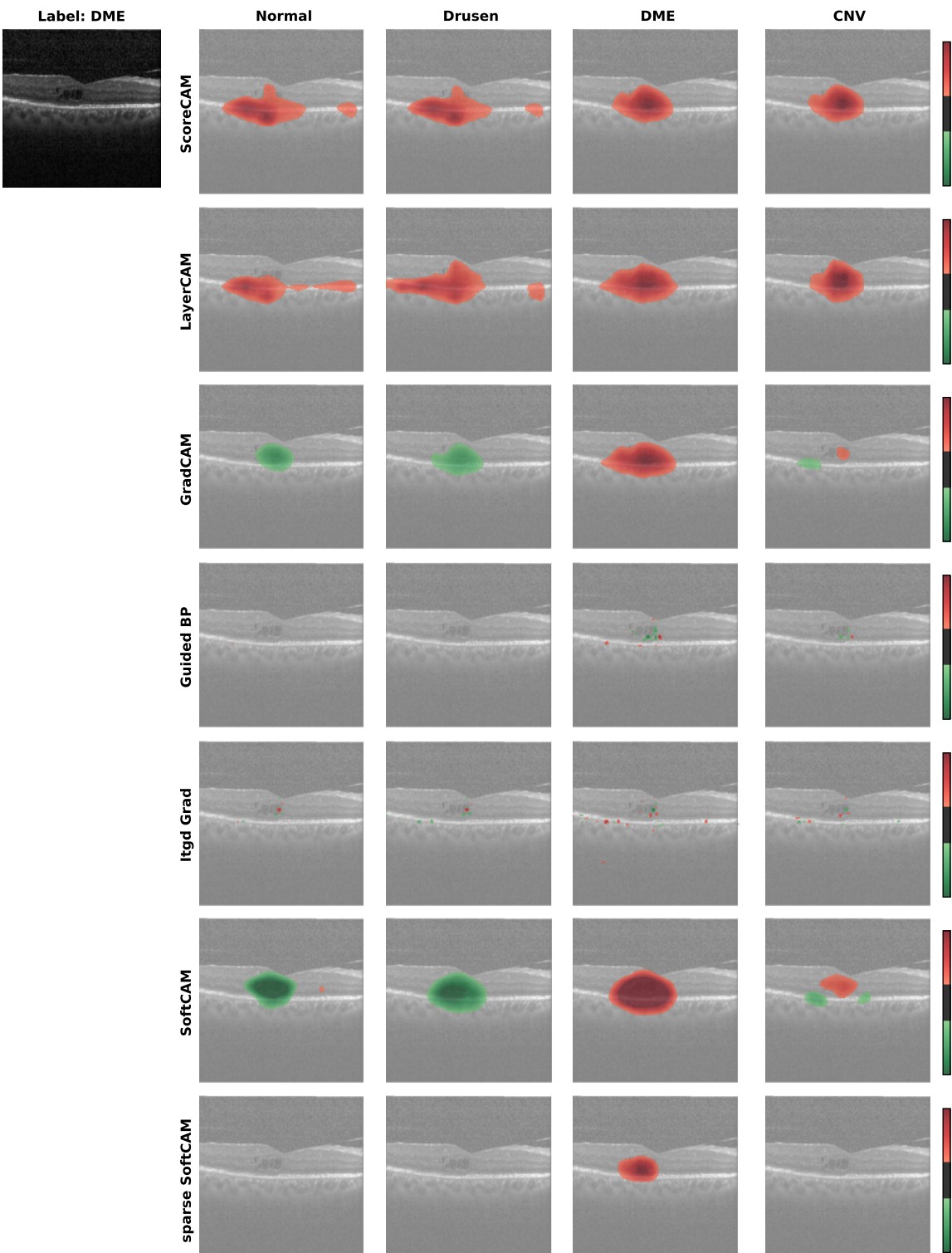

Figure 17: **Class-specific explanation with the ResNet backbone**. Sparse SoftCAM provides more precise class-specific localization than unregularized SoftCAM and leading post-hoc methods, highlighting clinically relevant regions.

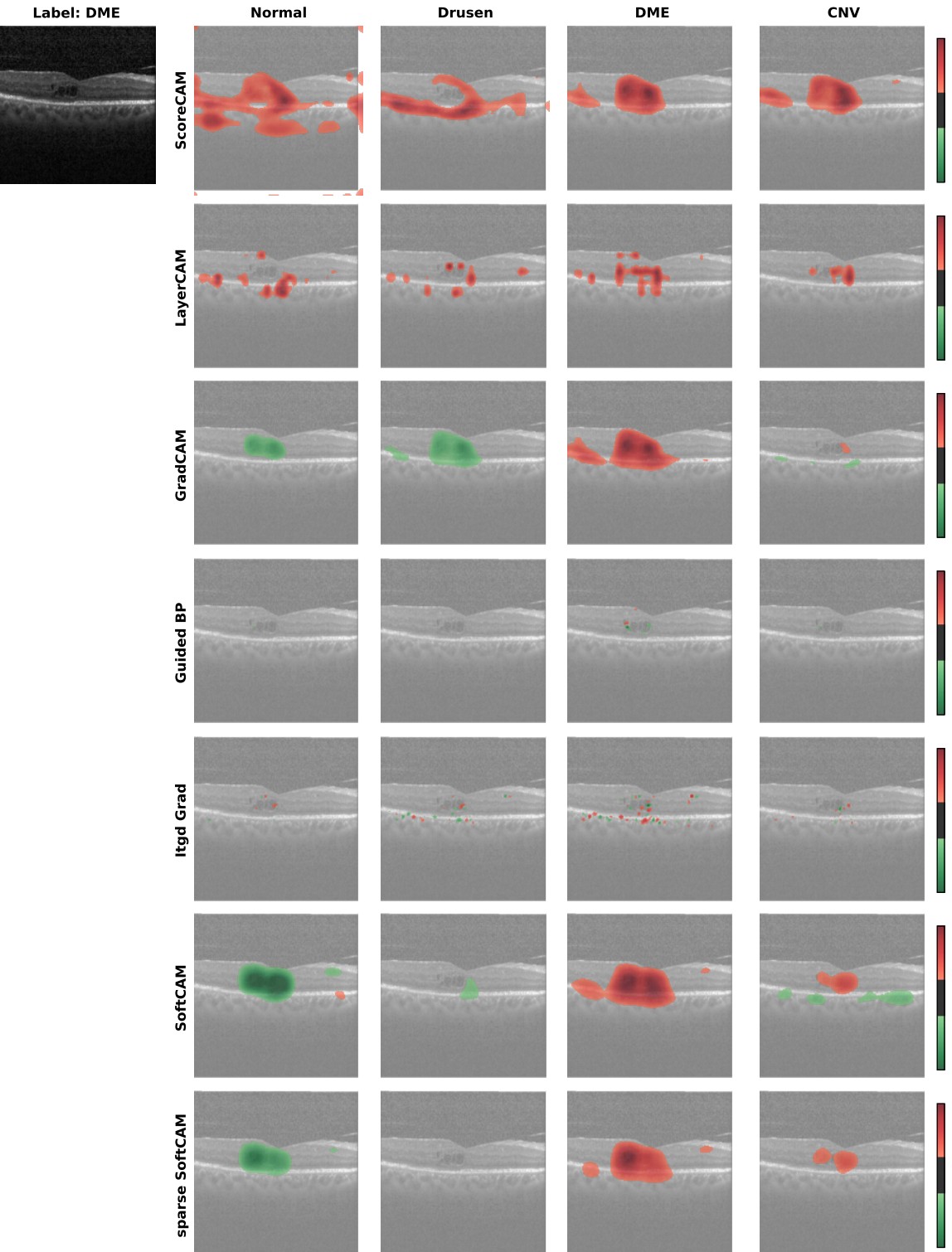

Figure 18: **Class-specific explanation with the VGG backbone**.

