# OpenReview forum: "SoftCAM: Making black box models self-explainable for medical image analysis"
_MIDL.io/2026/Conference — MIDL 2026 Poster_

### Official Review · Reviewer_TBYV · 2026-01-07

**Confidence:** 5
**Preliminary Rating:** 4
**Final Rating:** 5

**Summary:**

This paper proposes Soft-CAM, a self-explainable method designed to avoid post-hoc attribution. By replacing the GAP and FC layers with a 1×1 convolutional layer and applying regularization constraints, it directly generates saliency maps alongside predictions during training. The authors conduct a detailed comparative evaluation against several widely used post-hoc attribution methods on three datasets, demonstrating the superior interpretability of the proposed approach.

**Strengths:**

Soft-CAM enables CNNs to produce saliency maps simultaneously with predictions, thereby circumventing the unreliability associated with post-hoc attribution methods.
The methodology is clearly articulated, experimental details are thoroughly provided, and the supplementary material offers valuable extensions to the main content.
The study validates its findings using two network architectures and three datasets, and includes comprehensive comparisons with multiple interpretability methods across several evaluation metrics.

**Weaknesses:**

The proposed method has certain limitations regarding its applicability across diverse deep learning model architectures, and it inherently shares the same resolution restrictions as CAM-based methods in terms of saliency map granularity.
The comparative analysis lacks comparison with some recently proposed self-explainable methods.

**Detailed Comments:**

1. Although the proposed method is relatively straightforward, the comprehensive performance evaluation convincingly demonstrates that the method is more reliable than post-hoc attribution approaches. It is easy to adopt and technically sound, representing a solid contribution.
2. The use of an ElasticNet penalty to encourage evidence maps to minimize false negatives while promoting sparsity ensures a reliable and reasonable training process for Soft-CAM.
3. Some repetitive phrasing appears in the Method section on page 4 (e.g., “Unlike CAM…” and “In contrast to classical CAM-based methods…”). More concise wording would improve the manuscript's clarity.

**Justification Of Final Rating:**

I have revised my rating to Strong Accept, as the authors have addressed all of my questions. The methodological contributions and the detailed, rigorous experiments presented in the paper are commendable.

**Justification Of The Preliminary Rating:**

This paper presents Soft-CAM, a self-explainable method that avoids post-hoc attribution and provides thorough validation. The proposed approach is clear and effective, and I consider it a novel contribution. However, given certain methodological limitations, I am inclined towards a weak accept.

**Questions To Address In The Rebuttal:**

1. In the regularization design, the Lasso penalty encourages sparse evidence maps. Could this lead to suboptimal interpretability performance on medical images containing large lesion areas?
2. Several recent self-explainable methods for medical imaging have been proposed (e.g., https://arxiv.org/abs/2507.16761). Could the authors further clarify Soft-CAM’s advantages over such approaches—for instance, by discussing the limitations of the cited method or supplementing the paper with a comparative analysis?

---

> ### Author Response · Authors · 2026-01-24
> **Potential limitations compared to post-hoc methods, self-explainable models, and regularization design**
>
> Thank you for your helpful comments, particularly regarding clarification of the regularization design and comparisons with other self-explainable methods. We appreciate your acknowledgment of the clarity of the methodology, the comprehensive experimental setup involving two network architectures and three datasets, the extensive comparison with multiple posthoc interpretability methods and evaluation metrics, and the supplementary material, which provides valuable extensions to the main content.
>
> **[Regularization design. The Lasso penalty encourages sparse evidence maps. Could this lead to suboptimal interpretability performance on medical images containing large lesion areas?]**
>
> Thank you for highlighting this point. Indeed, because Lasso regularization encourages sparse evidence maps, it can lead to suboptimal interpretability performance on medical images that contain large lesion areas. This observation motivates our introduction of ElasticNet (L1/L2) regularization, whose balance can be empirically selected. For tasks involving small, localized lesions—such as diabetic retinopathy detection—sparse (Lasso) regularization is most appropriate. In contrast, for tasks with larger lesion regions, such as pneumonia detection in chest X-rays, ElasticNet (Lasso + Ridge) regularization is more suitable. We revised the manuscript in Sec. 4.4 to further emphasize this point.
>
> For example, as shown in Fig. 4, enforcing sparsity through a Lasso penalty leads to suboptimal activation precision on chest X-ray images, where regions of interest are typically large. In this setting, excessive sparsity increases false negatives by suppressing activations within relevant regions. Incorporating a Ridge penalty promotes denser activations, thereby improving activation sensitivity and overall interpretability. Future work could address principled ways to select the L1/L2 trade off in a data dependent manner.
>
> **[Limitations regarding applicability across diverse deep learning model architectures and restriction in terms of saliency map granularity]**
>
> The method was initially developed for classical CNN architectures; however, as already acknowledged in the discussion, future work will explore extensions to other deep learning models, such as Vision Transformers (ViTs). Additional directions, also already highlighted in the discussion section, include improving saliency map granularity and extending the approach to other tasks, such as weakly supervised segmentation. We revised the discussion section to further emphasize this point.
>
> **[Comparison with self-explainable methods]**
>
> Compared to existing self-explainable models, such as prototype-based approaches [2] and the B-cos network [1], SoftCAM introduces no additional computational overhead and remains straightforward and easy to implement. In contrast, it has been reported that B-cos transformations introduce significant computational overhead, increasing training and inference time by up to 60% compared to baseline models of the same size [1], while prototype-based models are often complex, difficult to train, and lack straightforward interpretability [2, 3]. While a comprehensive comparison with self-explainable methods is beyond the scope of this work, we will update the manuscript by including BcosNet in the related works and extend the respective discussion section.
>
> Thank you again for taking the time to review our paper and for your helpful feedback, including your recognition of the solidity of the contribution and your identification of some repetitive phrasing. We hope that these responses address your questions, allow you to increase your score and we remain open to further suggestions for improving the paper.
>
> **References**
> - [1] Böhle et al. "B-cos networks: Alignment is all we need for interpretability."
> - [2] Chen et al. “This looks like that: deep learning for interpretable image recognition”
> - [3] Djoumessi et al. “This actually looks like that: Proto-bagnets for local and global interpretability-by-design"

---

> > ### Comment · Reviewer_TBYV · 2026-01-28
> >
> > Thank you for the clarifications and for thoughtfully addressing my comments. The explanations provided effectively clarify the key aspects of the study, and the additional details incorporated into the revised manuscript further strengthen its rigor and clarity. All of my questions have been adequately addressed by the authors.

---

### Official Review · Reviewer_rQ9K · 2026-01-08

**Confidence:** 5
**Preliminary Rating:** 3
**Final Rating:** 4

**Summary:**

The paper proposes Soft-Cam, a simple modification of deep classifiers to make the network intrinsically explainable. For that, the authors propose to remove the fully connected layer commonly used at the end of the classification network and replace it by a 1x1 convolution with  "number of classes" output channels. Then, the spatial dimension is reduced to 1 with an average pooling and a softmax is used to make the decision. The maps produced by the 1x1 convolution are used to explain the network decision (attribution).
The authors also use a regularization on these maps based on ElasticNet (L1 + L2 losses) during training.

The method is evaluated on 3 datasets/tasks (Diabetic Retinopathy, Retinal OCT and Chest X-Ray). with two networks (Resnet50 and VGG16). The method is evaluated on classification performances and explanations (precision and sensitivity). The authors compared themselves to ScoreCam, LayerCam, GardCam, Guided back-propagation and Integrated Gradient.

The classification performances are comparable to the original network while the precision of the explanations is higher than state-of-the-art. The results are less convincing on the sensitivity.

**Strengths:**

- The attributions are obtained by simply looking at the maps after the 1x1 convolution.
- The method is simple and the code is provided so it can be easily reproduced.
- The experiments are performed on several datasets and networks.

**Weaknesses:**

- The proposed method is very similar to CAM (Learning deep features for discriminative localization, Zhou et al). The authors should compared themselves to CAM for each experiments on explainability.
- The proposed method has the same resolution problem as CAM, GradCam, etc., except that it cannot be applied to previous layers to avoid it. The improve the resolution, the network has to been reduced.
- The authors only show the results for the L1+L2 regularization loss for Chest X-Ray experiments. Why not on previous experiences (OCT and Fundus) that are more extensive?
- I do not understand how the $\lambda$ parameter was set (Figure 9): best accuracy for Fundus but not for OCT...
- Occlusion for sensitivity metric may lead to out-of-distribution images .
- A lot of important information (multi-class for example) is on the supplementary material (which is very long, by the way).

**Detailed Comments:**

/

**Justification Of Final Rating:**

The authors answered my questions. The method is interesting as it is simple and the code is provided so it can be easily reproduced. However, it requires a specific classification head and has poor resolution.

**Justification Of The Preliminary Rating:**

The method is interesting but the a comparison to CAM is necessary as it is very similar. Some details and explanations should also be added about the ElasticNet regularization and the $\lambda$ parameter setting.

**Questions To Address In The Rebuttal:**

- Adding a comparison to CAM
- Adding results with L1+L2 regularizations for Table 1 and Figure 2 experiments.
- Explanation on the $\lambda$ setting

---

> ### Author Response · Authors · 2026-01-24
> **Adding missing results (comparison with CAM, L2 regularization) and explaination of the $\lambda$ setting**
>
> Thank you for your helpful comments, particularly regarding the comparison with CAM, which is the closest posthoc method to SoftCAM, and for the suggestion to move some relevant information from the appendix to the main paper. We appreciate your acknowledgment of the method’s simplicity, as well as the comprehensive experimental setup involving multiple datasets and network architectures.
>
> **[Adding comparison to CAM]**
>
> Although CAM was one of the first class-specific, visualization-based methods to explicitly link CNN predictions to spatial regions, it requires a specific network architecture with a global average pooling (GAP) layer followed by a single linear classifier, as in ResNet-style architectures. As a result, CAM cannot be directly applied to standard VGG networks, which rely on multiple fully connected layers rather than GAP. Subsequent posthoc methods, such as GradCAM, were proposed to generalize and extend CAM to a wider range of architectures. While CAM is not expected to provide improved performance, we nevertheless include CAM results for the ResNet model in our paper for completeness (Fig. 2) and will also add the quantitative results to the relevant appendix .
>
> **[Adding results with L1+L2 regularizations for Table 1 and Figure 2 experiments]**
>
> The choice of L1, L2, or combined regularization is inherently task dependent. In our experiments, the fundus and OCT datasets contain highly localized and sparse ground-truth annotations, making Lasso regularization ($\lambda_2=0$ in Eq. 4) particularly effective. In contrast, chest X-ray annotations typically cover larger disease regions, for which Lasso regularization is generally less suitable and denser regularization schemes—such as Ridge regularization ($\lambda_1=0$ in Eq. 4 ) or combinations of L1 and L2—are more appropriate. We therefore added L2 results for Table 1 and Figure 2 as requested.
>
> Obtaining comparable results for fundus and OCT datasets, however, would require an additional extensive grid search over L2 regularization strengths, since this involves retraining the model to select appropriate hyperparameters. Given the limited rebuttal timeframe, we conducted a restricted grid search over a small set of \lambda_2 values for both fundus and OCT datasets and report the classification results of the best-performing configuration in Table 1, selected based on strong predictive performance and qualitatively meaningful visual explanations on a subset of annotated images. As we are still conducting experiments on the multi-class task, the results (Table 1) will be updated accordingly once they are finished.
>
> **[Explanation on the $\lambda$ setting]**
>
> We selected $\lambda$ such that classification accuracy and AUC/Kappa were not strongly diminished and visually checked on a few annotated example images. We have clarified this selection procedure in the revised text (Sec. 2.3, and Sec. 4.1). Further work should investigate more principled approaches for setting the $\lambda$ parameters.
>
> **[Occlusion for sensitivity metric may lead to out-of-distribution images]**
>
> Thank you for highlighting this point. Indeed, occlusion can create out-of-distribution inputs [6], which may invalidate sensitivity-based explanation evaluations. In such cases, performance degradation under occlusion may reflect distribution shift rather than the faithfulness. This is a well known and important limitation of occlusion-based metrics. We will explicitly acknowledge this issue in the discussion section, particularly in light of our findings showing discrepancies between sensitivity-based measures (occlusion) and other explainability metrics, despite sensitivity being a widely used and established approach for evaluating explanations.
>
> **[Some important information on the supplementary material]**
>
> We moved some of them into the main paper, notably key parts of the multi-class evaluation.
>
> **[Low resolution of the feature map]**
>
> Thank you for pointing out this limitation. As already acknowledged in the discussion, the resulting low-resolution feature maps limit spatial granularity and lead to coarse-grained explanations. In our case, the ElasticNet constraints help improve localization; however, future work could focus on increasing feature map resolution to enable more fine-grained explanations.
>
> Thank you again for taking the time to review our paper and for your helpful feedback, including the suggestion to add comparisons with CAM, report classification results for the L2-regularized models with a clear explanation of the \lambda settings, and move relevant information to the main manuscript. We hope that these responses address your questions and allow you to increase your score, and we remain open to further suggestions for improving the paper.
>
> Reference
> - [6] Hase et al. " The out-of-distribution problem in explainability and search methods for feature importance explanations."

---

> > ### Author Response · Authors · 2026-01-30
> >
> > Dear Reviewer rQ9K,
> >
> > did you have a chance to review our reply? It would be great to hear about your feedback, so we can clarify any remaining questions.
> >
> > Best
> > The Authors

---

> > > ### Comment · Reviewer_rQ9K · 2026-01-30
> > > **CAM**
> > >
> > > Thank you for your answer. Have you managed to obtain quantitative results for CAM?

---

> > ### Author Response · Authors · 2026-01-30
> > **Quantitative results for CAM and L2 regularization on the binary tasks.**
> >
> > Thank you once again for highlighting the comparison with CAM. Quantitative results show that, in terms of localization precision and faithfulness, CAM often performs comparably to some posthoc methods, notably ScoreCAM and LayerCAM; however, it consistently underperforms the SoftCAM variants, particularly the sparse variant (obtained with L1 regularization).
> >
> > As an illustration, we report below the quantitative results-- including CAM and L2 (ridge)--for top-15 localization precision and top-30 sensitivity. Sensitivity is quantified using the relative Area Under the Deletion Curve (AUDC), where lower values indicate greater faithfulness—that is, a larger decrease in the model’s confidence when the most relevant patches are removed. For localization precision, higher values indicate better alignment between saliency maps and ground-truth annotations.
> >
> > |   Fundus dataset  | top-15 Precision | top-30 Sensitivity |
> > |---------------------|------------------|--------------------|
> > | CAM                 | 0.255            | 0.639              |
> > | ScoreCAM            | 0.216            | 0.672              |
> > | LayerCAM            | 0.246            | 0.654              |
> > | GradCAM             | 0.373            | 0.639              |
> > | Guided Backprop     | 0.384            | **0.573**              |
> > | IntegratedGradients | 0.340            | 0.625              |
> > | SoftCAM             | 0.395            | 0.69               |
> > | ridge SoftCAM       | 0.373            | 0.697              |
> > | sparse SoftCAM      | **0.524**            | 0.687              |
> >
> > |  Retina OCT dataset | top-15 Precision | top-30 Sensitivity |
> > |---------------------|------------------|--------------------|
> > | CAM                 | 0.12             | 0.731              |
> > | ScoreCAM            | 0.122            | 0.731              |
> > | LayerCAM            | 0.11             | 0.737              |
> > | GradCAM             | 0.152            | 0.731              |
> > | Guided Backprop     | 0.359            | 0.681              |
> > | IntegratedGradients | 0.303            | 0.697              |
> > | SoftCAM             | 0.462            | 0.613              |
> > | ridge SoftCAM       | 0.526            | 0.557              |
> > | sparse SoftCAM      | **0.863**            | **0.315**              |
> >
> >
> > | RSNA dataset  | top-15 Precision | top-30 Sensitivity |
> > |---------------------|------------------|--------------------|
> > | CAM                 | 0.745            | 0.95               |
> > | ScoreCAM            | 0.746            | 0.97               |
> > | LayerCAM            | 0.722            | 0.957              |
> > | GradCAM             | 0.751            | 0.95               |
> > | Guided Backprop     | 0.598            | 0.978              |
> > | IntegratedGradients | 0.563            | 0.981              |
> > | SoftCAM             | 0.738            | 0.921              |
> > | ridge SoftCAM       | 0.696            | **0.897**              |
> > | sparse SoftCAM      | **0.783**            | 0.924              |
> >
> > As expected, CAM does not yield improved quantitative performance. Nevertheless, we will report these quantitative results in the relevant appendix for completeness.
> >
> > We hope that these results address your questions, and we remain open to further suggestions for improving the paper.

---

### Official Review · Reviewer_PMer · 2026-01-10

**Confidence:** 5
**Preliminary Rating:** 5

**Summary:**

This work introduces SoftCAM, an approach that makes CNNs inherently interpretable simply by removing the global average pooling layer and replacing it with a 1x1 convolution.

The extensive experimental setup and the results show that SoftCAM allows maintaining the classification performance while outperforming state-of-the-art post-hoc interpretability methods in several explainability metrics.

**Strengths:**

The paper is very well written and structured. It has been an absolute pleasure reading it! The captions are also perfect!

The proposed method is simple, making it easy to implement in other CNN architectures.

The method is tested on several CNNs and datasets, showing its generalizability and versatility.

The experimental setup is very well designed, with experiments on classification performance and several interpretability metrics, especially localization precision (with clinical validation!) and faithfulness, and not simply presenting the qualitative results, as unfortunately many papers in the XAI field do. The faithfulness computation is particularly well designed, as the authors only compute faithfulness for correctly predicted instances on all networks tested, thus ensuring a fair comparison.

The authors also propose a new metric, activation sensitivity.

Another very important point is that authors specifically evaluate quantitatively the maps produced for healthy images, something that is rarely seen in the literature on XAI.

The discussion regarding the difference between human- and model-aligned metrics is very interesting and relevant, since the community tends to assume the models classify similarly to how humans classify the images.

Overall a very very important paper for the XAI community!

**Weaknesses:**

One weakness, or rather, limitation of SoftCAM is the fact that it requires retraining the underlying CNN when compared to post-hoc methods that are plug-and-play.

Additionally, the authors do not compare their method to any other inherently interpretable method. I understand that many inherently interpretable methods are very custom and implementing them would be extremely costly, but one recent method is fairly easy to implement (there is code available). I am referring to B-cos networks [1] which have been proposed for natural images and later have been evaluated on medical images [2].


[1] Böhle, Moritz, Mario Fritz, and Bernt Schiele. "B-cos networks: Alignment is all we need for interpretability." Proceedings of the IEEE/CVF Conference on Computer Vision and Pattern Recognition. 2022.

[2] Rio-Torto, Isabel, et al. "On the Suitability of B-cos Networks for the Medical Domain." 2024 IEEE International Symposium on Biomedical Imaging (ISBI). IEEE, 2024.

**Detailed Comments:**

I did not notice any grammatical errors or typos.

The only thing would be that in the text SoftCAM is written differently from in the title, where it appears Soft-CAM.

The authors could more explicitly list the introduction of a novel explainability metric as another contribution of the paper.

**Justification Of The Preliminary Rating:**

The paper is very well written and structured.

The authors propose a novel attribution method that could even be applied outside the medical domain. The validation of the method, especially in terms of faithfulness, shows it is faithful to the model decisions.

The authors also propose a novel interpretability metric.

Overall, it is an extremely relevant work for the XAI community.

**Questions To Address In The Rebuttal:**

Comparison with, at least, one inherently interpretable method (e.g. b-cos networks). And, of course, including it in the related works.

---

> ### Author Response · Authors · 2026-01-24
> **Potential limitations and comparison with inherently  interpretable methods**
>
> Thank you for your helpful and very positive comments, especially regarding comparisons with interpretable models. We appreciate your acknowledgment of the paper’s quality and structure, as well as the comprehensive experimental setup evaluating multiple interpretability metrics, including quantitative analysis of explanation maps for healthy images and the introduction of a new evaluation metric.
>
> **[Comparison to inherently interpretable methods]**
>
> The main motivation for this work comes from well-known limitations of posthoc explanation methods that are currently widely adopted, particularly in medical image analysis. We argue that such posthoc explanations should be avoided in the medical domain in favor of self-explainable alternatives. In this work, we propose SoftCAM, which maintains high predictive performance while offering substantially improved explainability. Compared to existing self-explainable models, such as prototype-based approaches and the B-cos network, SoftCAM introduces no additional computational overhead and remains straightforward and easy to implement. In contrast, it has been reported that B-cos transformations introduce significant computational overhead, increasing training and inference time by up to 60% compared to baseline models of the same size [1], while prototype-based models are often complex, difficult to train, and lack straightforward interpretability [2, 3]. While a comprehensive comparison with self-explainable methods is beyond the scope of this work, we will update the manuscript by including BcosNet in the related works and extend the respective discussion section.
>
> **[One weakness, or rather, limitation of SoftCAM is the fact that it requires retraining the underlying CNN when compared to post-hoc methods that are plug-and-play]**
>
> Indeed, we advocate the use of inherently-interpretable architectures from the start, as it has been shown that posthoc explanations methods are not informative when applied to complex decisions functions [4]. Nevertheless, SoftCAM can also be applied posthoc to blackbox models, as in [5] (see App. A), by removing the GAP layer and replacing the fully connected classifier with convolution layers that share weights with the main classifier during inference. A thorough analysis of this extension, however, is beyond the scope of the present work.
>
> Thank you again for taking the time to review our paper and for your helpful feedback, including pointing out the self-explainable BcosNet, the inconsistent spelling of “SoftCAM” and suggesting that we could more clearly emphasize the introduction of the novel explainability metric, activation precision, as an additional contribution that was previously overlooked. We hope that these responses address your questions, and we remain open to further suggestions for improving the paper.
>
> References
> - [1] Böhle et al. "B-cos networks: Alignment is all we need for interpretability."
> - [2] Chen et al. “This looks like that: deep learning for interpretable image recognition”
> - [3] Djoumessi et al. “This actually looks like that: Proto-bagnets for local and global interpretability-by-design"
> - [4] Günther et al. “Informative Post-Hoc Explanations Only Exist for Simple Functions”

---

### Author Rebuttal · Authors · 2026-01-24

**Rebuttal:**

We thank all reviewers for their helpful comments and for acknowledging the clarity of the methodology and the comprehensive experimental setup. We have addressed the reviewers' concerns in the respective response sections for each reviewer, with changes highlighted in colored text in the revised manuscript.

Concretely:
- We added text to clarify the selection of the $\lambda$ hyperparameters (Reviewer rQ9K, Reviewer TBYV).
- We included the self-explainaible model BcosNet in the related work and discussion section (Reviewer PMer, Reviewer TBYV).
- We added the multi-class performance results, previously located in the appendix, to Table 1; included a comparison with CAM in Figure 2; and added L2 regularization results to Table 1 and Figure 2. We also moved the multi-class setting from the appendix to the main paper, added two new figures (Figs. 5 and 6) that were previously in the appendix, and completely rewrote the multi-class section to reflect these changes (Reviewer rQ9K).
- As the multi-class experiments with L2 regularization are still running, we will update Table 1 with the corresponding results and revise Figure 6 by replacing the post-hoc method (Integrated Gradients) with the L2-regularized model. We will also update the related figures and tables in the appendix to reflect the new results for CAM and L2 regularization in both binary and multi-class settings.

We sincerely appreciate all feedback provided by the reviewers once more.

**Supporting Material:**

/attachment/e70cb3de0d6cc0c9ec4de2ac95225ff0a941d15c.pdf

---

### Comment · Area_Chair_pvNx · 2026-01-29
**Final Rating**

Dear Reviewers,
we appreciate your active participation in the discussion.
Please consider the rebuttals of the authors, as well as the revised manuscripts, and set your final rating by clicking "Edit"->"Official Review" by  February 1st 2026 (23:59 AoE).
best regards

---

### Meta-Review · Area_Chair_pvNx · 2026-02-03

**Recommendation:** Accept (Poster)
**Confidence:** 5

**Metareview:**

The authors addressed most of the reviewers concerns, and the reviewers agree that despite some methodological limitations, the method is novel and a valuable contribution to the field. I therefore recommend an "accept". I would like to thank the reviewers and the authors for their work!

---

### Decision · Program_Chairs · 2026-02-13

Accept (Poster)